# Revisiting Uncertainty:
# On Evidential Learning for Partially Relevant Video Retrieval

**Jun Li** [1 2]  **Peifeng Lai** [3]  **Xuhang Lou** [3]  **Jinpeng Wang** [3]  **Yuting Wang**
**Ke Chen** [2]  **Yaowei Wang** [3 2]  **Shu-tao Xia** [1 2]

## Abstract

Partially relevant video retrieval aims to retrieve untrimmed videos using text queries that describe only partial content. However, the inherent asymmetry between brief queries and rich video content inevitably introduces *uncertainty* into the retrieval process. In this setting, vague queries often induce *semantic ambiguity* across videos, a challenge that is further exacerbated by the *sparse temporal supervision* within videos, which fails to provide sufficient matching evidence. To address this, we propose *Holmes*, a hierarchical evidential learning framework that aggregates multi-granular cross-modal evidence to quantify and model uncertainty explicitly. At the **inter-video** level, similarity scores are interpreted as evidential support and modeled via a Dirichlet distribution. Based on the proposed three-fold principle, we perform fine-grained query identification, which then guides query-adaptive calibrated learning. At the **intra-video** level, to accumulate denser evidence, we formulate a soft query-clip alignment via flexible optimal transport with an adaptive dustbin, which alleviates sparse temporal supervision while suppressing spurious local responses. Extensive experiments demonstrate that *Holmes* outperforms state-of-the-art methods. Code is released at ICML26-Holmes.

## 1. Introduction

With the rapid growth of online videos, Text-to-Video Retrieval (T2VR) (Lan et al., 2025) has received increasing attention. While most T2VR methods are designed for pre-trimmed videos with fully descriptive queries, they may deviate from real-world settings, where videos are generally untrimmed and queries correspond only to partial segments. Motivated by this discrepancy, *Partially Relevant Video Retrieval* (PRVR) (Dong et al., 2022a) was proposed for retrieving untrimmed videos given partially relevant queries. However, this task faces a fundamental challenge rooted in data asymmetry: compared with rich video content, text queries are often brief and vague, inevitably introducing inherent uncertainty into the retrieval process.

Modeling this uncertainty involves two compounded challenges that exist at both inter- and intra-video levels: (**i**) **Inter-video Semantic Ambiguity.** Owing to the inherent uncertainty, queries often exhibit fine-grained heterogeneity. As illustrated in Figure 1(a-d), beyond ideal *precise queries*, real-world data contains *under-determined queries* (insufficient semantics yielding low confidence), and *polysemous queries* (ambiguous semantics matching multiple candidates). Most prior methods ignore this heterogeneity (Wang et al., 2024b) or handle it implicitly via regularization (Zhang et al., 2025a; Moon et al., 2025b). While ARL (Cho et al., 2025) attempts to detect ambiguous video-text pairs, it lacks a mechanism to explicitly quantify the specific type of uncertainty. (**ii**) **Intra-video Sparse Supervision.** The inter-video ambiguity is also exacerbated by the lack of reliable evidence within the video. The prevalent multi-instance learning (MIL) paradigm supervises only the single best-matching clip, leading to *sparse supervision*. As shown in Figure 1(e), this supervisory imbalance provides limited training signals and prevents the model from learning a holistic representation, making it vulnerable to spurious local noise. While distilling cues from CLIP (Radford et al., 2021) can alleviate sparsity (Dong et al., 2023), such frame-level fragments fail to capture the temporal dynamics required to confirm a video's relevance, thereby fueling the uncertainty at the inter-video level and resulting suboptimal results.

To address these challenges, we argue that the model needs to go beyond deterministic similarity scores and should evaluate the uncertainty of its predictions. In this paper, we propose *Holmes*, a unified evidential learning framework that operates at dual levels to disentangle and model uncer-

[1]Tsinghua Shenzhen International Graduate School, Tsinghua University, Shenzhen, China [2]Peng Cheng Laboratory, Shenzhen, China [3]Harbin Institute of Technology, Shenzhen, China. Correspondence to: Jinpeng Wang <wangjp26@gmail.com>.

*Proceedings of the 43rd International Conference on Machine Learning*, Seoul, South Korea. PMLR 306, 2026. Copyright 2026 by the author(s).

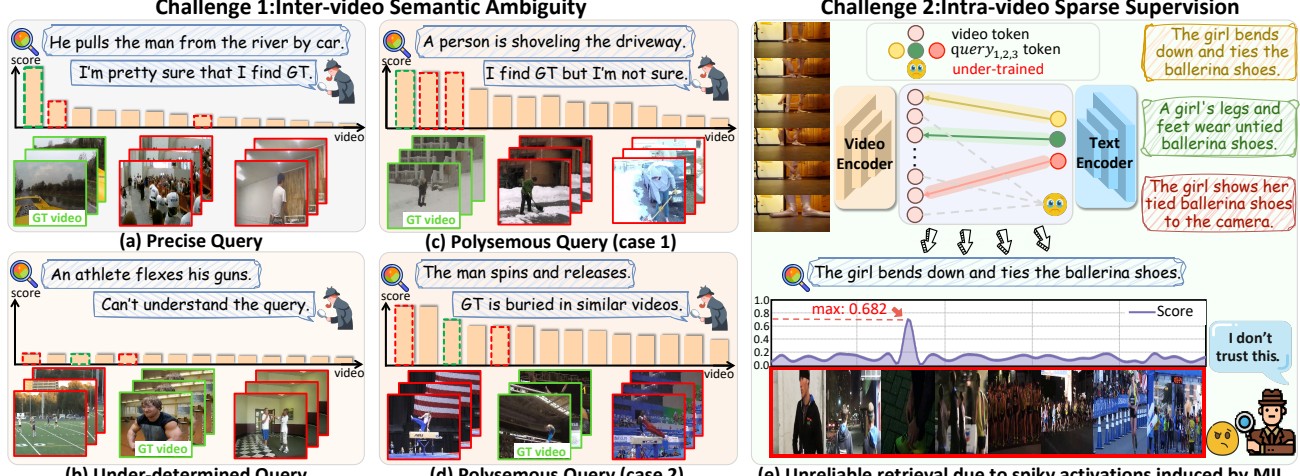

*Figure 1.* (a) *Precise queries* produce a clear response margin, with the ground-truth video distinctly outperforming all candidates. (b) *Under-determined queries* are poorly interpreted, resulting in weak and non-discriminative similarity responses. (c-d) *Polysemous queries* are semantically ambiguous and correspond to multiple plausible candidates, necessitating *more consideration* to account for the associated uncertainty. (e) Optimizing only the closest pair, MIL yields *sparse supervision* and under-trained clips, making the model vulnerable to local noise from a globally irrelevant "running" video, causing spurious spiky activations and unreliable retrieval.

tainty. First, to handle **inter-video uncertainty**, we treat cross-modal similarities as evidence parameterized by a Dirichlet distribution. This allows us to move beyond simple matching scores and perform a *diagnosis* of the query type. We devise an Uncertainty Guided Identification (UGI) strategy based on a three-fold principle, namely *epistemic uncertainty*, *label consistency*, and *aleatoric uncertainty*, to categorize queries and apply query-adaptive label calibration. Second, to resolve the **intra-video sparsity** that fuels uncertainty, we seek to accumulate denser evidence. We formulate a Flexible Optimal Transport (FOT) scheme to establish soft alignments between the query and multiple relevant clips. To ensure robustness against noise, we integrate a *dustbin bucket* to absorb irrelevant clips, thereby providing more comprehensive and reliable temporal supervision.

Extensive experiments on ActivityNet Captions (Krishna et al., 2017), Charades-STA (Gao et al., 2017), and TVR (Lei et al., 2020) demonstrate that *Holmes* achieves state-of-the-art performance. Beyond standard retrieval metrics, our analysis reveals that *Holmes* effectively interprets query ambiguity and suppresses spurious MIL activations.

Our main contributions are summarized as follows:

- ***Conceptual***: We provide a fresh perspective on PRVR by revisiting the problem through the lens of uncertainty, explicitly linking inter-video semantic ambiguity with intra-video sparse supervision.
- ***Methodological***: We propose *Holmes*, a dual-level evidential learning framework. It features a three-fold principle for fine-grained query identification and label calibration, coupled with Flexible Optimal Transport to enable dense and robust temporal alignment.

- ***Empirical***: *Holmes* achieves significant improvements on three benchmarks. Detailed ablation studies further validate its capability in quantifying query uncertainty and effectively suppressing spurious local noise.

## 2. Related Works

**Partially Relevant Video Retrieval**  Unlike traditional Text-to-Video Retrieval (T2VR) methods (Reddy et al., 2025; Lan et al., 2025; Ko et al., 2025), which retrieve fully relevant pre-trimmed videos, Partially Relevant Video Retrieval (PRVR) (Li et al., 2026; Jun et al., 2025; Yin et al., 2024; Song et al., 2025; Jiang et al., 2023), first introduced by MS-SL (Dong et al., 2022a), targets untrimmed videos where only partial segments match the query. Video Corpus Moment Retrieval (VCMR) (Zhang et al., 2025b; Chen et al., 2023c) is another widely studied retrieval task that aims to localize specific moments within a large video corpus. PRVR is similar to the first stage of VCMR, as both conduct coarse video-level retrieval. But VCMR requires moment-level annotations, which hinders scalability.

Existing PRVR methods largely ignore semantic ambiguity and primarily focus on clip modeling. MS-SL (Dong et al., 2022a; Chen et al., 2025b) constructs redundant clip embeddings via a sliding window scheme, whereas Proto-PRVR (Moon et al., 2025a) accelerates retrieval by generating a small set of prototypes. GMMFormer (Wang et al., 2024b;a) implicitly models clips through Gaussian attention. While RAL (Zhang et al., 2025a) leverages probabilistic embeddings and MSC-PRVR (Moon et al., 2025b) alleviates semantic collapse through auxiliary training losses to implicitly model query ambiguity, ARL (Cho et al., 2025)

explicitly identifies ambiguous query-video pairs. Nevertheless, these methods fail to distinguish fine-grained query categories and overlook the sparse supervision induced by multiple instance learning, which leads to suboptimal retrieval performance. We propose *Holmes*, which identifies query heterogeneity through inter-video evidential learning and establishes soft query-clip associations via intra-video evidential learning, providing denser supervisory signals.

**Uncertainty Learning** Despite their success, deep neural networks remain largely deterministic and lack explicit mechanisms for uncertainty quantification. Early attempts to estimate uncertainty primarily relied on Bayesian Neural Networks (BNNs) (Franchi et al., 2023) and Monte Carlo Dropout (Gal & Ghahramani, 2016). More recently, Evidential Deep Learning (EDL) (Sensoy et al., 2018), grounded in Dempster-Shafer Theory (DST) (Shafer, 1992) and Subjective Logic (SL) (Jøsang, 2016), has attracted increasing attention by explicitly modeling uncertainty through second-order probability distributions and has been successfully applied to various tasks, *e.g.*, multi-view classification (Han et al., 2022; Xu et al., 2024), noisy correspondence multimodal retrieval (Li et al., 2025a; Qin et al., 2025; 2022; Zha et al., 2024; 2025), temporal action localization (Chen et al., 2023a;b), open-set action recognition (Bao et al., 2021; Zhao et al., 2023) and zero-shot learning (Huang et al., 2024), with ongoing efforts (Deng et al., 2023; Chen et al., 2025a; Yoon & Kim, 2025) further improving the capability of uncertainty estimation of EDL. Motivated by these advances, we design a hierarchical evidential learning framework that aggregates multi-granularity cross-modal evidence to jointly quantify retrieval uncertainty in PRVR.

## 3. Method

### 3.1. Problem Statement and Overview

Partially Relevant Video Retrieval aims to retrieve videos that contain moments relevant to a given text query. Each video consists of multiple candidate moments, while each query corresponds to a specific moment within its relevant video. In this paper, we propose *Holmes*, a hierarchical evidential learning framework for addressing query ambiguity in PRVR at both inter-video and intra-video levels, as shown in Figure 1(a). We first introduces three components below.

**Text Query Representation** A text query is first encoded by a pre-trained RoBERTa (Liu et al., 2019), followed by a projection layer and a Transformer (Vaswani et al., 2017) to obtain contextualized representations, which are aggregated into the final $q \in \mathbb{R}^d$ similar to MS-SL (Dong et al., 2022a).

**Video Representation** Given an untrimmed video, we first extract features using a pre-trained CNN. Following prior arts (Wang et al., 2024b), we adopt a dual-branch architecture to capture multi-scale representations. In the

frame-scale branch, $M_f$ sampled frames are projected to a $d$-dimensional space via a fully connected layer, followed by frame encoder to obtain $\boldsymbol{V_f} = \{\boldsymbol{f}_i\}_{i=1}^{M_f} \in \mathbb{R}^{M_f \times d}$. In the clip-scale branch, the video is sparsely segmented into $M_c$ clips, which are first projected and then processed by clip encoder to produce $\boldsymbol{V_c} = \{\boldsymbol{c}_i\}_{i=1}^{M_c} \in \mathbb{R}^{M_c \times d}$.

**Similarity Computation** To compute the similarity of a text–video pair $(T, V)$, we first obtain $\boldsymbol{q}$, $\boldsymbol{V_f}$, and $\boldsymbol{V_c}$ and compute frame- and clip-scale similarity scores:

$$\begin{aligned} s^f(T, V) &= \max\{\cos(\boldsymbol{q}, \boldsymbol{f}_1), ..., \cos(\boldsymbol{q}, \boldsymbol{f}_{M_f})\}, \\ s^c(T, V) &= \max\{\cos(\boldsymbol{q}, \boldsymbol{c}_1), ..., \cos(\boldsymbol{q}, \boldsymbol{c}_{M_c})\}. \end{aligned} \quad (1)$$

We then compute the overall text-video similarity score:

$$s(T, V) = \alpha_f s^f(T, V) + \alpha_c s^c(T, V), \quad (2)$$

where $\alpha_f, \alpha_c \in [0, 1]$ and $\alpha_f + \alpha_c = 1$. Videos are finally retrieved and ranked according to Equation (2).

### 3.2. Inter-video Evidential Learning

This section models query ambiguity via query uncertainty estimation with inter-video evidential learning. We first introduce the three-fold principle for query identification (Figure 2(a)) and describe the framework using the frame-scale branch, as both branches share similar pipelines.

**Three-fold Principle** For query $\boldsymbol{q}_i$ and $K$ videos, we first compute the frame-scale similarity $s_{ij}^f$ with the $j$-th video (Equation (1)), denoted as $s_{ij}$ for convenience. Evidence is derived via similarities following (Sensoy et al., 2018):

$$e_{ij} = \exp\left(\tanh\left(s_{ij}/\tau\right)\right), \quad \boldsymbol{e}_i = [e_{i1}, \ldots, e_{iK}], \quad (3)$$

where $\tau = 0.1$ by default. According to Dempster-Shafer Theory (DST) (Shafer, 1992), epistemic uncertainty can be quantified by evidence and reflect how strongly the data support the association between a query and candidates. We first map the evidence $e_i$ to Dirichlet parameters:

$$\boldsymbol{\alpha}_i = [\alpha_{i1}, \ldots, \alpha_{iK}], \quad \alpha_{ij} = e_{ij} + 1. \quad (4)$$

**Definition 3.1. Epistemic Uncertainty.** For query $\boldsymbol{q}_i$, epistemic uncertainty $u_i$ and belief mass $b_{ij}$ are defined as

$$u_i = \frac{K}{S_i}, \quad b_{ij} = \frac{\alpha_{ij} - 1}{S_i}, \quad S_i = \sum_{j=1}^{K} \alpha_{ij}, \quad (5)$$

with $u_i + \sum_j b_{ij} = 1$. Here, $S_i$ denotes the Dirichlet distribution strength and $\boldsymbol{b}_i = [b_{i1}, \ldots, b_{iK}]$ represents the subjective opinion derived from $\boldsymbol{\alpha}_i$.

High $u_i$ captures under-determined queries by reflecting limited evidence and weak query comprehension, whereas low $u_i$ does not ensure precise retrieval. Formally,

**Theorem 3.2.** *A low epistemic uncertainty $u_i$ does not imply that the highest belief is assigned to the annotated label $y_i$.*

$$\boldsymbol{q}_i \text{ with low } u_i \not\Rightarrow \arg\max \boldsymbol{b}_i = \arg\max \boldsymbol{y}_i. \quad (6)$$

See Appendix B.2 for the proof. $\boldsymbol{y}_i = [y_{i1}, \ldots, y_{iK}]$ is the one-hot ground-truth label. This shows that epistemic uncertainty cannot ensure belief concentration on the annotated video, motivating our label consistency principle.

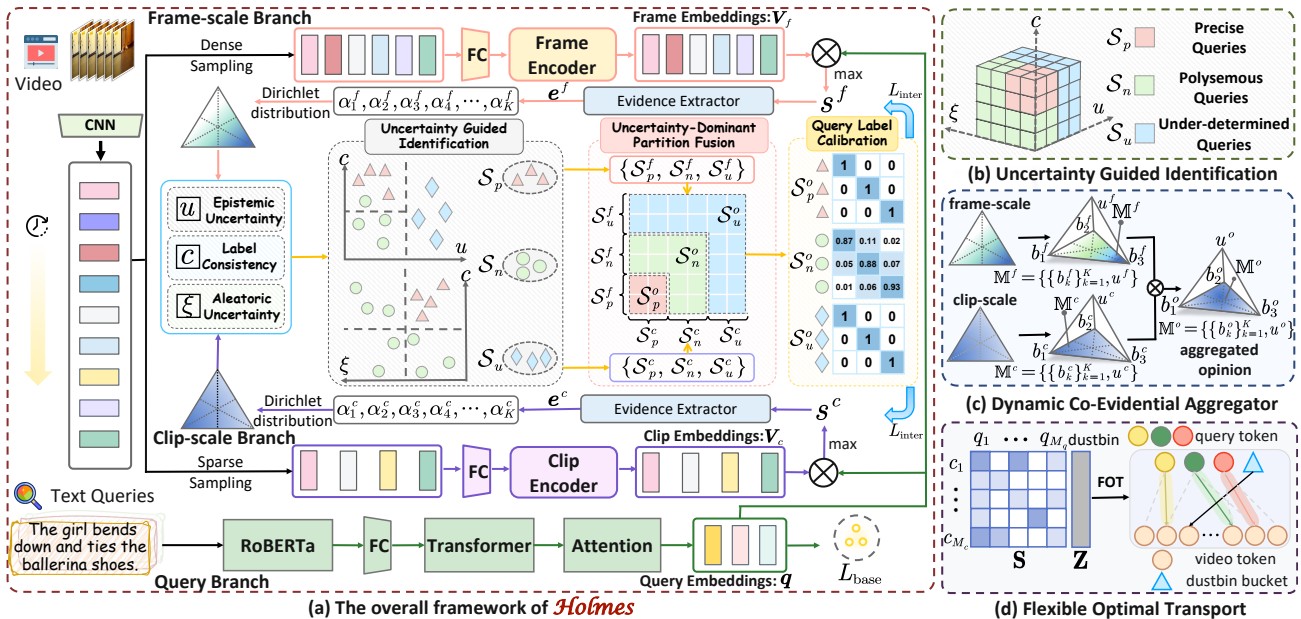

*Figure 2.* Overview of $\mathcal{H}olmes$. **(a)** The query branch produces the query embedding $\boldsymbol{q}$, while the frame- and clip-scale branches extract video representations $\boldsymbol{V}_f$ and $\boldsymbol{V}_c$, yielding similarity scores $S_f$ and $S_c$. The candidate similarities are converted into evidence vectors and modeled with Dirichlet distributions to estimate the three-fold principle, based on which Uncertainty-Guided Identification partitions queries at each scale into precise ($\mathcal{S}_p^f$, $\mathcal{S}_p^c$), polysemous ($\mathcal{S}_n^f$, $\mathcal{S}_n^c$), and under-determined ($\mathcal{S}_u^f$, $\mathcal{S}_u^c$) subsets. The dual-scale partitions are then integrated by Uncertainty-Dominant Partition Fusion to obtain the final results $\mathcal{S}_p^o$, $\mathcal{S}_n^o$, and $\mathcal{S}_u^o$. Finally, after query-specific label calibration, the refined labels supervise inter-video evidential learning via $L_{\text{inter}}$. **(b)** Uncertainty-Guided Identification partitions queries in a three-dimensional latent space through coarse separation on the $c$–$u$ plane followed by refinement along $\xi$. **(c)** Dynamic Co-Evidential Aggregator fuses dual-scale evidential opinions in a parameter-free manner. **(d)** Flexible Optimal Transport employs an *dustbin bucket* to absorb noisy clips, establishing reliable soft query-clip assignments as label evidence for intra-video evidential learning.

**Definition 3.3. Label Consistency.** For query $\boldsymbol{q}_i$, the label consistency $c_i$ is defined as:

$$c_i = \max(0, \boldsymbol{s}_i \cdot \boldsymbol{y}_i), \qquad (7)$$

where $\boldsymbol{s}_i = [s_{i1}, \ldots, s_{iK}]$ denotes similarity scores.

While low $c_i$ indicates poor alignment with the ground-truth, high $c_i$ do not indicate query precision. Formally,

**Proposition 3.4.** *A high label consistency $c_i$ does not imply that the query is precise.*

$$\boldsymbol{q}_i \text{ with high } c_i \not\Rightarrow \boldsymbol{q}_i \text{ is precise query.} \qquad (8)$$

*Proof.* Figure 1(c) shows that a high response from the ground-truth video does not imply a precise match, as other videos may exhibit similarly high similarity. $\qquad\square$

To distinguish precise and polysemous queries, we introduce aleatoric uncertainty (Ulmer et al., 2023) measured by the expected entropy of $\text{Dir}(\boldsymbol{\alpha}_i)$ in Equation (9). Low entropy indicates concentration of probability mass on a single video, suggesting precise queries, while high entropy reflects a more uniform distribution, indicating polysemous queries.

**Definition 3.5. Aleatoric Uncertainty.** For query $\boldsymbol{q}_i$, aleatoric uncertainty $\xi_i$ is defined as

$$\mathbb{E}_{\mathbf{p}\sim\text{Dir}(\boldsymbol{\alpha}_i)}[H(\mathbf{p})]=\sum_{k=1}^{K}\frac{\alpha_{ik}}{S_i}\left(\psi(S_i+1)-\psi(\alpha_{ik}+1)\right), \quad (9)$$

where $\psi$ denotes the digamma function, defined as $\psi(x) = \frac{d}{dx}\log\Gamma(x)$ and $H$ denotes the Shannon entropy. Please refer to Appendix B.3 for the full derivation.

**Uncertainty Guided Identification** Based on the above principles, queries can be categorized into three types: (i) under-determined queries with high epistemic uncertainty, $\mathcal{S}_u$; (ii) initially precise queries with low epistemic uncertainty and high label consistency, $\tilde{\mathcal{S}}_p$; (iii) initially polysemous queries with low epistemic uncertainty and low label consistency, $\tilde{\mathcal{S}}_n$, which can be formally expressed as

$$\begin{cases} \mathcal{S}_u = \{q_i \mid u_i > \beta_u\}, \\ \tilde{\mathcal{S}}_p = \{q_i \mid u_i \le \beta_u, c_i \ge \beta_p\}, \\ \tilde{\mathcal{S}}_n = \{q_i \mid u_i \le \beta_u, c_i < \beta_p\}. \end{cases} \qquad (10)$$

The thresholds $\beta_u$ and $\beta_p$ are adaptively determined as

$$\beta_u = \min(u^{tp}, 1 - \beta), \quad \beta_p = \max(\beta, c^{tp}), \qquad (11)$$

where $u^{tp}=\max_{i\in\mathcal{S}^{tp}} u_i$, $c^{tp} = \min_{i\in\mathcal{S}^{tp}} c_i$ and $\beta = 0.3$. $\mathcal{S}^{tp} = \{i \mid \arg\max(\boldsymbol{s}_i) = \arg\max(\boldsymbol{y}_i)\}$ denotes the correctly matched query–video pairs. $\tilde{\mathcal{S}}_n$ is often treated as noisy pairs in noisy correspondence learning. However, we interpret $\tilde{\mathcal{S}}_n$ as reflecting query ambiguity, where ambiguous semantic may weaken responses to ground-truth videos. Proposition 3.4 shows that $\tilde{\mathcal{S}}_p$ still contain polysemous queries. We further refine $\tilde{\mathcal{S}}_p$ by comparing $\xi$, yielding the final precise queries $\mathcal{S}_p$ and polysemous queries $\mathcal{S}_n$:

$$\begin{cases} \mathcal{S}_p = \{q_i \in \tilde{\mathcal{S}}_p \mid \xi_i < \text{median}(\{\xi_j\}_{q_j \in \tilde{\mathcal{S}}_p})\}, \\ \mathcal{S}_n = (\tilde{\mathcal{S}}_p \cup \tilde{\mathcal{S}}_n) \setminus \mathcal{S}_p. \end{cases} \quad (12)$$

**Uncertainty-Dominant Partition Fusion** Based on Equations (10) and (12), the frame-scale and clip-scale branches independently yield query partitions $\{\mathcal{S}_u^f, \mathcal{S}_p^f, \mathcal{S}_n^f\}$ and $\{\mathcal{S}_u^c, \mathcal{S}_p^c, \mathcal{S}_n^c\}$. However, these two branches may yield inconsistent predictions for the same query, motivating a principled mechanism to reconcile discrepancies.

**Definition 3.6. Uncertainty Dominance Ordering.** An *uncertainty dominance ordering* over query categories is

$$\mathcal{S}_p \prec \mathcal{S}_n \prec \mathcal{S}_u, \quad (13)$$

which induces a total order according to increasing semantic ambiguity and uncertainty.

This ordering is motivated by the observation that precise queries are semantically reliable, polysemous queries admit multiple candidates, whereas under-determined queries lack sufficient evidence for confident matching. Accordingly, cross-branch conflicts are resolved by selecting the identification with higher uncertainty dominance:

$$\begin{cases} \mathcal{S}_p^o = \mathcal{S}_p^f \cap \mathcal{S}_p^c, \\ \mathcal{S}_n^o = (\mathcal{S}_n^f \cap \mathcal{S}_p^c) \cup (\mathcal{S}_p^f \cap \mathcal{S}_n^c) \cup (\mathcal{S}_n^f \cap \mathcal{S}_n^c), \\ \mathcal{S}_u^o = (\mathcal{S}_u^f \cap \mathcal{S}_u^c) \cup (\mathcal{S}_u^f \cap (\mathcal{S}_p^c \cup \mathcal{S}_n^c)) \\ \qquad \cup (\mathcal{S}_u^c \cap (\mathcal{S}_p^f \cup \mathcal{S}_n^f)). \end{cases} \quad (14)$$

**Query Label Calibration** Given the final $\mathcal{S}_p^o$, $\mathcal{S}_n^o$, and $\mathcal{S}_u^o$, we perform query-specific label calibration. For precise queries in $\mathcal{S}_p^o$, the original labels are preserved. For under-determined queries in $\mathcal{S}_u^o$, which remain under-fitted due to insufficient semantic evidence, we retain the original labels to encourage learning. In contrast, for polysemous queries in $\mathcal{S}_n^o$, one-hot supervision overlooks semantic ambiguity, potentially over-penalizing semantically related samples and introducing misleading supervision. We recalibrate their labels by using model-estimated similarities to yield soft alignment probabilities that capture latent relevance. Given the $i$-th query with label $y_i$, the refined label $\hat{y}_i$ is:

$$\hat{y}_i = \begin{cases} y_i, & i \in \mathcal{S}_p^o \cup \mathcal{S}_u^o, \\ (1-\gamma)y_i + \dfrac{\gamma}{2}\left(\sigma(s_i^f) + \sigma(s_i^c)\right), & i \in \mathcal{S}_n^o, \end{cases} \quad (15)$$

where $\sigma(\cdot)$ denotes the softmax operation and $\gamma = 0.2$. $s_i^f$ and $s_i^c$ are the frame-scale and clip-scale similarity scores.

**Dynamic Co-Evidential Aggregator** As is shown in Figure 2 (c), for a query $q$, we first obtain the frame-scale and clip-scale evidential opinions $\mathbb{M}^f = \left\{\{b_k^f\}_{k=1}^K, u^f\right\}$ $\mathbb{M}^c = \left\{\{b_k^c\}_{k=1}^K, u^c\right\}$ according to Equations (4) and (5). The two branch opinions are then fused using the Dempster–Shafer Theory combination rule, yielding $\mathbb{M}^o = \left\{\{b_k^o\}_{k=1}^K, u^o\right\}$:

$$b_k^o = \frac{1}{1-\delta}(b_k^f b_k^c + b_k^f u^c + b_k^c u^f), \ u^o = \frac{1}{1-\delta}u^f u^c, \quad (16)$$

where $\delta = \sum_{i \neq j} b_i^f b_j^c$ quantifies conflict between the two opinions. This unified opinion could capture the holistic

uncertainty of the query and serve as a global supervisory signal for coordinating the two branches.

With multi-scale evidential opinions, query categorization, and label calibration established, we introduce the evidential learning objective. We formulate cross-modal retrieval as a $K$-way classification problem and optimize *Holmes* by minimizing the least-squares loss between the Dirichlet-based query probability $\boldsymbol{p}_i$ and the calibrated label $\hat{\boldsymbol{y}}_i$, defined as:

$$\begin{aligned} L_U(\boldsymbol{\alpha}_i, \hat{\boldsymbol{y}}_i) &= \int \|\hat{\boldsymbol{y}}_i - \boldsymbol{p}_i\|_2^2 \frac{1}{B(\boldsymbol{\alpha}_i)} \prod_{j=1}^K p_{ij}^{\alpha_{ij}-1} d\boldsymbol{p}_i \\ &= \sum_{j=1}^K \left(\hat{y}_{ij} - \frac{\alpha_{ij}}{S_i}\right)^2 + \frac{\alpha_{ij}(S_i - \alpha_{ij})}{S_i^2(S_i+1)}. \end{aligned} \quad (17)$$

Additional details are provided in Appendix B.4. Equation (17) enforces higher evidence for matched pairs than for mismatched ones, thereby preserving reliable uncertainty learning. Finally, for the $i$-th query, the overall objective of inter-video evidential learning is defined as:

$$L_{\text{inter}} = L_U^f\left(\boldsymbol{\alpha}_i^f, \hat{\boldsymbol{y}}_i\right) + L_U^c(\boldsymbol{\alpha}_i^c, \hat{\boldsymbol{y}}_i) + L_U^o(\boldsymbol{\alpha}_i^o, \hat{\boldsymbol{y}}_i), \quad (18)$$

where $\boldsymbol{\alpha}_i^f$, $\boldsymbol{\alpha}_i^c$, and $\boldsymbol{\alpha}_i^o$ are the Dirichlet parameters from the frame-scale, clip-scale, and aggregated evidential opinions.

### 3.3. Intra-video Evidential Learning

To mitigate the sparse supervision inherent in MIL, we employ optimal transport to establish a soft query-to-clip alignment within each video. Specifically, given clip embeddings $\{c_i\} \in \mathbb{R}^{M_c \times d}$ and corresponding query embeddings $\{q_j\} \in \mathbb{R}^{M_q \times d}$, where $M_q$ denotes the number of queries. Let $\mathbf{S} \in \mathbb{R}^{M_c \times M_q}$ denote the clip-query similarity matrix. $\mathbf{Q}$ is the transport assignment. The objective of OT is:

$$\max_{\mathbf{Q} \in \mathcal{Q}} \quad \langle \mathbf{Q}, \mathbf{S} \rangle + \varepsilon H(\mathbf{Q})$$

$$\text{s.t. } \mathcal{Q} = \left\{ \mathbf{Q} \in \mathbb{R}_+^{M_c \times M_q} \mid \mathbf{Q}\mathbf{1}_{M_q} = \boldsymbol{\mu}, \mathbf{Q}^\top \mathbf{1}_{M_c} = \boldsymbol{\nu} \right\}, \quad (19)$$

where $\mathbf{1}_{M_q}$ represents the vector of ones in dimension $M_q$, $\boldsymbol{\mu} \in \mathbb{R}^{M_c}$ and $\boldsymbol{\nu} \in \mathbb{R}^{M_q}$. $H(\mathbf{Q})$ is an entropy regularizer and $\varepsilon$ controls its smoothness. We obtain the optimal $\mathbf{Q}^*$ via the Sinkhorn algorithm (Cuturi, 2013).

**Flexible Optimal Transport** Optimal transport enforces full source-target mapping, which is ill-suited to PRVR where queries correspond to only partial clips and many clips are noisy or redundant. Motivated by (Sarlin et al., 2020; Lin et al., 2024), we introduce an adaptive dustbin bucket to filter such clips, as shown in Figure 2(d). The bucket is a constant column vector $z$, set to the bottom 30% percentile of $\mathbf{S}$ and appended to it:

$$\bar{\mathbf{S}} = [\mathbf{S} \mid \mathbf{Z}], \quad \mathbf{Z} = z\mathbf{1}_{M_c \times 1}. \quad (20)$$

By replacing Equation (19) with Equation (20), we obtain the final assignment by dropping the dustbins, $\bar{\mathbf{Q}}^* = \bar{\mathbf{Q}}_{1:M_c, 1:M_q}^*$, which provides a soft one-to-many matching.

*Table 1.* **Retrieval performance comparison on three standard datasets.** Models are ranked by ascending SumR on ActivityNet Captions. State-of-the-art results are shown in bold, while "–" indicates unavailable entries.

| Model | ActivityNet Captions | | | | | Charades-STA | | | | | TVR | | | | |
|---|---|---|---|---|---|---|---|---|---|---|---|---|---|---|---|
| | R@1 | R@5 | R@10 | R@100 | SumR | R@1 | R@5 | R@10 | R@100 | SumR | R@1 | R@5 | R@10 | R@100 | SumR |
| *Text-to-Video Retrieval (T2VR)* | | | | | | | | | | | | | | | |
| RIVRL (Dong et al., 2022b) | 5.2 | 18.0 | 28.2 | 66.4 | 117.8 | 1.6 | 5.6 | 9.4 | 37.7 | 54.3 | 9.4 | 23.4 | 32.2 | 70.6 | 135.6 |
| CLIP4Clip (Luo et al., 2022) | 5.9 | 19.3 | 30.4 | 71.6 | 127.3 | 1.8 | 6.5 | 10.9 | 44.2 | 63.4 | 9.9 | 24.3 | 34.3 | 72.5 | 141.0 |
| Cap4Video (Wu et al., 2023) | 6.3 | 20.4 | 30.9 | 72.6 | 130.2 | 1.9 | 6.7 | 11.3 | 45.0 | 65.0 | 10.3 | 26.4 | 36.8 | 74.0 | 147.5 |
| *Video Corpus Moment Retrieval (VCMR)* | | | | | | | | | | | | | | | |
| ReLoCLNet (Zhang et al., 2021) | 5.7 | 18.9 | 30.0 | 72.0 | 126.6 | 1.2 | 5.4 | 10.0 | 45.6 | 62.3 | 10.0 | 26.5 | 37.3 | 81.3 | 155.1 |
| CONQUER (Hou et al., 2021) | 6.5 | 20.4 | 31.8 | 74.3 | 133.1 | 1.8 | 6.3 | 10.3 | 47.5 | 66.0 | 11.0 | 28.9 | 39.6 | 81.3 | 160.8 |
| JSG (Chen et al., 2023c) | 6.8 | 22.7 | 34.8 | 76.1 | 140.5 | 2.4 | 7.7 | 12.8 | 49.8 | 72.7 | - | - | - | - | - |
| *Partially Relevant Video Retrieval (PRVR)* | | | | | | | | | | | | | | | |
| MS-SL (Dong et al., 2022a) | 7.1 | 22.5 | 34.7 | 75.8 | 140.1 | 1.8 | 7.1 | 11.8 | 47.7 | 68.4 | 13.5 | 32.1 | 43.4 | 83.4 | 172.4 |
| MS-SL++ (Chen et al., 2025b) | 7.0 | 23.1 | 35.2 | 75.8 | 141.1 | 1.8 | 7.6 | 12.0 | 48.4 | 69.7 | 13.6 | 33.1 | 44.2 | 83.5 | 174.5 |
| PEAN (Jiang et al., 2023) | 7.4 | 23.0 | 35.5 | 75.9 | 141.8 | **2.7** | 8.1 | 13.5 | 50.3 | 74.7 | 13.5 | 32.8 | 44.1 | 83.9 | 174.2 |
| LH (Fang et al., 2024) | 7.4 | 23.5 | 35.8 | 75.8 | 142.4 | 2.1 | 7.5 | 12.9 | 50.1 | 72.7 | 13.2 | 33.2 | 44.4 | 85.5 | 176.3 |
| BGM-Net (Yin et al., 2024) | 7.2 | 23.8 | 36.0 | 76.9 | 143.9 | 1.9 | 7.4 | 12.2 | 50.1 | 71.6 | 14.1 | 34.7 | 45.9 | 85.2 | 179.9 |
| GMMFormer (Wang et al., 2024b) | 8.3 | 24.9 | 36.7 | 76.1 | 146.0 | 2.1 | 7.8 | 12.5 | 50.6 | 72.9 | 13.9 | 33.3 | 44.5 | 84.9 | 176.6 |
| MamFusion (Ying et al., 2025) | 8.0 | 25.4 | 37.2 | 76.8 | 147.4 | 2.0 | 8.8 | 14.2 | 51.5 | 76.5 | 14.2 | 33.9 | 44.9 | 84.5 | 177.5 |
| ProtoPRVR (Moon et al., 2025a) | 7.9 | 24.9 | 37.2 | 77.3 | 147.4 | - | - | - | - | - | 15.4 | 35.9 | 47.5 | 86.4 | 185.1 |
| DL-DKD (Dong et al., 2023) | 8.0 | 25.0 | 37.5 | 77.1 | 147.6 | - | - | - | - | - | 14.4 | 34.9 | 45.8 | 84.9 | 179.9 |
| ARL (Cho et al., 2025) | 8.3 | 24.6 | 37.4 | 78.0 | 148.3 | - | - | - | - | - | 15.6 | 36.3 | 47.7 | 86.3 | 185.9 |
| MGAKD (Zhang et al., 2025c) | 7.9 | 25.7 | 38.3 | 77.8 | 149.6 | - | - | - | - | - | 16.0 | 37.8 | 49.2 | 87.5 | 190.5 |
| DL-DKD++ (Dong et al., 2025) | 8.3 | 25.5 | 38.3 | 77.8 | 149.9 | 1.9 | 7.1 | 12.3 | 49.8 | 71.1 | 15.3 | 36.0 | 47.5 | 86.0 | 184.8 |
| GMMFormerV2 (Wang et al., 2024a) | 8.9 | 27.1 | 40.2 | 78.7 | 154.9 | 2.5 | 8.6 | 13.9 | 53.2 | 78.2 | 16.2 | 37.6 | 48.8 | 86.4 | 189.1 |
| HLFormer (Li et al., 2025b) | 8.7 | 27.1 | 40.1 | 79.0 | 154.9 | 2.6 | 8.5 | 13.7 | 54.0 | 78.7 | 15.7 | 37.1 | 48.5 | 86.4 | 187.7 |
| DreamPRVR (Li et al., 2026) | 8.7 | 27.5 | 40.3 | 79.5 | 156.1 | 2.6 | 8.7 | 14.5 | **54.2** | 80.0 | 17.4 | 39.0 | 50.4 | 86.2 | 193.1 |
| *Holmes* (ours) | **9.3** | **27.8** | **40.5** | **79.1** | **156.8** | 2.3 | **9.5** | **15.2** | 53.6 | **80.6** | **18.4** | **40.7** | **52.0** | **87.5** | **198.6** |

Treating each intra-video alignment as an $M_c$-way classification, we can follow Equations (3) and (4) to collect evidence and estimate the Dirichlet parameters. By viewing $\bar{\mathbf{Q}}^*$ as the calibrated label, we define the intra-video evidential learning objective $L_{\text{intra}}$ based on Equation (17). For each query, $L_{\text{intra}}$ encourages the accumulation of strong evidence for *multiple relevant* clips and weak evidence for irrelevant ones, thereby guiding the model to acquire denser and more comprehensive temporal supervision.

### 3.4. Model Optimization

Besides the $L_{\text{intra}}$ and $L_{\text{inter}}$ proposed by *Holmes*, following MS-SL (Dong et al., 2022a), we employ the standard similarity retrieval loss $L_{\text{sim}}$ and a query diversity $L_{\text{div}}$ (Wang et al., 2024b) to enhance retrieval performance. These two components form the basic training loss: $L_{\text{base}} = L_{\text{sim}} + L_{\text{div}}$.

Therefore, the aggregate loss is defined as:

$$L_{\text{agg}} = L_{\text{base}} + L_{\text{inter}} + L_{\text{intra}}. \tag{21}$$

Training follows a two-stage scheme: $L_{\text{base}}$ is applied throughout, the first stage serves as a warm-up, activating only $L_{\text{inter}}$ without query label recalibration to learn initial text-video matching. In the second stage, all objectives are jointly optimized, including $L_{\text{intra}}$ and label recalibration.

## 4. Experiments

### 4.1. Experimental Setup

**Datasets**   We conduct experiments on three benchmark datasets. **(i)** ActivityNet Captions (Krishna et al., 2017) con-

*Table 2.* **Training and Inference Efficiency.** Inference time measures feature extraction for 4430 videos, while retrieval time is the total time for encoding, similarity computation, and ranking over 15,753 queries on ActivityNet Captions evaluation set.

| Model | Train time/epoch (ms) | Model parameters (M) | Inference time (ms) | Retrieval time (ms) | SumR |
|---|---|---|---|---|---|
| GMMFormer | 38079 | 12.85 | 5473 | 15016 | 146.0 |
| HLFormer | 52866 | 28.43 | 6547 | 17001 | 154.9 |
| GMMFormerV2 | 62458 | 30.79 | 6578 | 17263 | 154.9 |
| *Holmes* | 52658 | 30.79 | 6556 | 17058 | 156.8 |

tains approximately 20K YouTube videos with an average duration of 118 seconds, each annotated with an average of 3.7 moments paired with textual descriptions. **(ii)** Charades-STA (Gao et al., 2017) comprises 6,670 videos with 16,128 sentence descriptions, averaging 2.4 moments per video. **(iii)** TV Show Retrieval (TVR) (Lei et al., 2020) consists of 21.8K video clips from six TV shows, each accompanied by five natural language descriptions. For fair comparison, we adopt the same data splits as in MS-SL (Dong et al., 2022a).

**Metrics**   We adopt rank-based metrics for evaluation, specifically $R@K (K = 1, 5, 10, 100)$. $R@K$ is defined as the percentage of queries where the ground-truth item appears within the top $K$ ranked results. We also report the Sum of Recalls (SumR) to show the holistic performance. All results are presented in percentages (%).

### 4.2. Implementation Details

**Data Pre-Processing**   For ActivityNet Captions and Charades-STA, video representations are obtained using the provided I3D features from Zhang et al. (2020) and

*Table 3.* Extensive ablation studies of *Holmes*. The best scores are marked in **bold**.

| ID | Model | ActivityNet Captions | | | | | Charades-STA | | | | | TVR | | | | |
|---|---|---|---|---|---|---|---|---|---|---|---|---|---|---|---|---|
| | | R@1 | R@5 | R@10 | R@100 | SumR | R@1 | R@5 | R@10 | R@100 | SumR | R@1 | R@5 | R@10 | R@100 | SumR |
| (0) | *Holmes* (ours) | **9.3** | **27.8** | **40.5** | **79.1** | **156.8** | 2.3 | **9.5** | **15.2** | **53.6** | **80.6** | **18.4** | **40.7** | **52.0** | 87.5 | **198.6** |
| *Efficacy of Multi-scale Inter-video Evidential Learning* | | | | | | | | | | | | | | | | |
| (1) | Frame-scale Only | 8.6 | 26.6 | 39.4 | 78.6 | 153.2 | 2.4 | 8.8 | 14.6 | 52.6 | 78.4 | 16.8 | 38.4 | 50.5 | 87.5 | 193.1 |
| (2) | Clip-scale Only | 8.5 | 26.6 | 39.8 | 78.6 | 153.6 | 2.4 | 8.6 | 14.4 | 52.6 | 78.0 | 17.1 | 38.8 | 50.6 | 87.6 | 194.1 |
| (3) | $w/o$ Aggregation scale | 8.7 | 27.1 | 39.8 | 78.7 | 154.4 | 2.3 | 9.2 | 14.9 | 52.6 | 79.1 | 17.3 | 40.2 | 51.5 | 87.7 | 196.7 |
| *Efficacy of various Partition Fusion Strategies* | | | | | | | | | | | | | | | | |
| (5) | Trust Frame-scale | 8.8 | 27.3 | 40.0 | 78.8 | 155.0 | 2.3 | 8.9 | 14.6 | 52.7 | 78.5 | 17.2 | 39.4 | 50.6 | 87.5 | 194.7 |
| (6) | Trust Clip-scale | 8.9 | 27.0 | 40.0 | 78.9 | 154.8 | 2.3 | 8.5 | 14.3 | 53.4 | 78.5 | 17.0 | 39.8 | 51.1 | 87.5 | 195.4 |
| (7) | $w/$ Confidence Dominance | 9.1 | 27.4 | 40.1 | 78.9 | 155.5 | 2.3 | 9.3 | 14.8 | 53.0 | 79.3 | 17.9 | 40.0 | 51.4 | 87.8 | 197.0 |
| *Efficacy of Query Label Calibration* | | | | | | | | | | | | | | | | |
| (8) | $w/o$ Calibration | 8.8 | 27.4 | 39.7 | 78.7 | 154.7 | 2.3 | 8.3 | 14.2 | 52.7 | 77.4 | 17.4 | 39.7 | 51.5 | 87.7 | 196.3 |
| (9) | $\tilde{S}_n$ Only | 9.1 | 27.6 | 40.1 | 78.9 | 155.6 | 2.3 | 9.0 | 15.0 | 53.1 | 79.4 | 17.7 | 40.4 | 51.4 | 87.8 | 197.3 |
| *Efficacy of Intra-video Evidential Learning* | | | | | | | | | | | | | | | | |
| (10) | $w/o\ L_{\text{intra}}$ | 8.7 | 27.1 | 39.7 | 78.6 | 154.0 | 2.1 | 8.7 | 14.4 | 52.4 | 77.7 | 17.3 | 39.2 | 50.7 | 87.7 | 194.9 |
| *Efficacy of Flexible Optimal Transport* | | | | | | | | | | | | | | | | |
| (11) | $w/$ softmax | 8.7 | 27.3 | 40.0 | 78.7 | 154.7 | 2.3 | 8.6 | 14.2 | 53.5 | 78.6 | 17.6 | 39.8 | 51.2 | 87.5 | 196.1 |
| (12) | $w/$ OT | 9.0 | 27.5 | 40.3 | 78.8 | 155.6 | 2.3 | 9.0 | 15.0 | 53.1 | 79.4 | 18.2 | 40.4 | 51.5 | 87.4 | 197.5 |

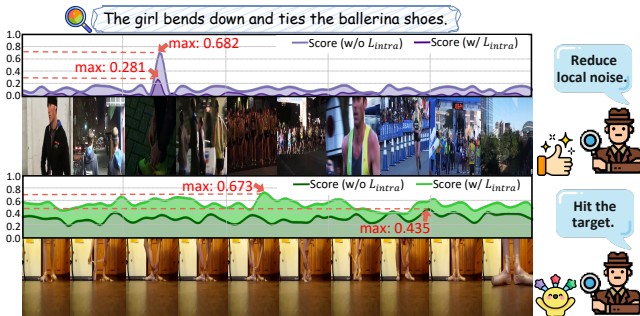

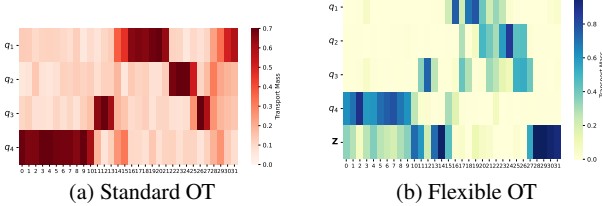

(a) Standard OT      (b) Flexible OT

*Figure 4.* Comparison of transport mass alignment between OT and FOT. $Z$ denotes the dustbin bucket, which is observed in the visualization to absorb the mass from the last several clips.

*Figure 3.* Using the same videos as in Figure 1 for comparison, removing $L_{\text{intra}}$ leads to retrieval failure, as the irrelevant video receives a higher response score than the ground-truth video (0.682 vs. 0.435). In contrast, incorporating $L_{\text{intra}}$ correctly ranks the ground-truth video above irrelevant ones (0.281 vs. 0.673).

Mun et al. (2020), respectively. Query representations are encoded as 1,024-dimensional RoBERTa features following MS-SL (Dong et al., 2022a). For TVR, we employ the 3,072-dimensional video features released by Lei et al. (2020), which combine frame-level ResNet152 (He et al., 2016) and segment-level I3D (Carreira & Zisserman, 2017) representations. Corresponding textual data is encoded as 768-dimensional RoBERTa features (Liu et al., 2019).

**Experimental Configurations** To maintain the same model size, the frame/clip encoder adopts 8 Gaussian attention blocks following (Wang et al., 2024a). The latent dimension $d = 384$. The model is implemented in PyTorch and trained on a single NVIDIA RTX 3080 Ti GPU for 100 epochs, with the first stage lasting 20 epochs and a batch size of 128. Other details are depicted in Appendix A.1.

### 4.3. Comparison with State-of-the arts

**Baselines** We select 15 representative PRVR baselines for comparison. In addition, we evaluate *Holmes* against other methods in T2VR and VCMR. For T2VR, we compare with RIVRL (Dong et al., 2022b), CLIP4Clip (Luo et al., 2022) and Cap4Video (Wu et al., 2023). For VCMR, the comparison includes ReLoCLNet (Zhang et al., 2021), CONQUER (Hou et al., 2021) and JSG (Chen et al., 2023c).

**Retrieval Performance** Table 1 summarizes the retrieval performance of different models across three benchmarks. PRVR-specific methods consistently outperform T2VR and VCMR approaches. By effectively modeling query ambiguity and providing reliable supervision for more video clips, *Holmes* achieves state-of-the-art performance. Specifically, it outperforms the strongest competitor, DreamPRVR(Li et al., 2025b), by 0.7 and 0.6 in SumR on ActivityNet Captions and Charades-STA, respectively, and further surpasses MGAKD(Zhang et al., 2025c) by 8.1 in SumR on TVR.

**Model Efficiency** We further evaluate the model efficiency in Table 2. All results are averaged over 5 runs under identical settings. Compared with similar-scale models, *Holmes* achieves better performance with comparable efficiency, indicating that hierarchical evidential learning incurs negligible overhead and demonstrates high efficiency.

### 4.4. Model Analyses

**Multi-scale Inter-video Evidential Learning**     We perform ablations to evaluate the multi-scale branch by (i) using only the frame- or clip-scale branch, training only with $L_U^f$ or $L_U^c$ and (ii) removing the Dynamic Co-Evidential Aggregator Module, *i.e.*, training $w/o\ L_U^o$. As shown in Table 3, single-branch learning and the absence of aggregation both lead to inferior performance, showing the necessity of multi-granularity inter-video evidential learning. It also suggests that cross-branch evidence aggregation could yield an overall uncertainty estimate that promotes optimization.

**Effects of Various Partition Fusion Strategies**     We compare three fusion strategies: (i) trusting single-branch, (ii) UDPF (default), which addresses inter-branch conflicts by favoring hypotheses with higher uncertainty and (iii) Confidence Dominance, which adopts the opposite fusion principle. As shown in Table 3, fusing dual-branch predictions yields more reliable assignments and improved accuracy. Compared to the conservative UDPF strategy, the more aggressive Confidence Dominance may induce over-confident yet incorrect associations, resulting in inferior performance.

**Efficacy of Query Label Calibration**     Comparing ID (0), ID (8), and ID (9) in Table 3 shows that uncalibrated baseline performs worst. We attribute this to its neglect of inherent semantic ambiguity, which indiscriminately pushes unpaired videos, over-penalizing semantically related samples and introducing misleading supervision. Meanwhile, the initial polysemous set $\tilde{S}_n$ fails to capture polysemous queries present in $\tilde{S}_p$, leading to suboptimal performance. These results further validate the importance of modeling query ambiguity and fine-grained query identification.

**Efficacy of Intra-video Evidential Learning**     ID (10) in Table 3 shows that removing $L_{\text{intra}}$ causes a notable drop in retrieval accuracy. We further visualize the temporal attention scores between queries and videos in Figure 3. With $L_{\text{intra}}$, the model exhibits suppressed responses to incorrect videos and reduced spurious local peaks, while producing higher similarity scores on relevant ones. By establishing soft one-to-many query-clip assignments and aggregating evidence over multiple relevant clips, it provides denser temporal supervision, enabling the model to mitigate spurious local noise and improve retrieval reliability and accuracy.

**Effect of Flexible Optimal Transport**     To evaluate the effect of Flexible Optimal Transport, we compare three variants: (i) softmax-based assignment, (ii) optimal transport (OT) and (iii) Flexible Optimal Transport (FOT, default). As shown in Table 3, softmax performs worst, while OT improves performance but fails to capture partial relevance in PRVR. In contrast, as illustrated in Figure 4, FOT with the proposed dustbin bucket absorbs noisy clips and mitigates irrelevant misalignments, achieving the best performance.

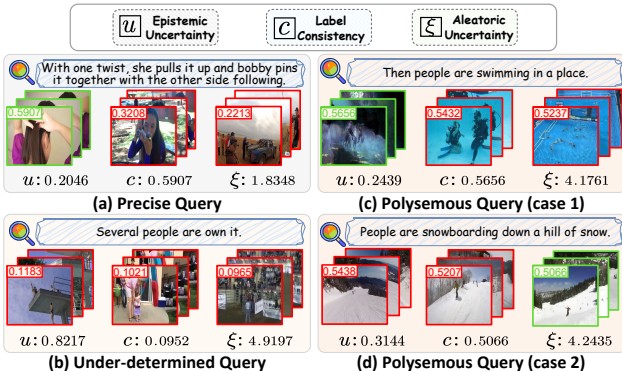

*Figure 5.* Qualitative examples of query identification. For each query, the top-3 retrieved videos are shown, with correct matches outlined in green and incorrect matches in red. Similarity scores are displayed at the upper-left corner of each video.

**Qualitative Query Identification Results**     Figure 5 presents representative examples demonstrating the effectiveness of our identification algorithm. Under-determined queries are clearly separated by $u$ and $c$, while precise and polysemous queries are mainly distinguished by $\xi$, consistent with our categorization criteria. We observe that, compared with relatively short polysemous queries without explicit referential cues, precise queries are generally longer and more spefic. Additionally, under-determined queries may exhibit grammatical or spelling errors (*e.g.*, misspellings such as own), which hinder semantic understanding. These observations also align well with real-world scenes and further support the validity of our approach.

## 5. Conclusions

In this paper, we presented *Holmes*, a hierarchical evidential learning framework designed to address the inherent uncertainty and supervisory sparsity in PRVR. To tackle query ambiguity at the inter-video level, *Holmes* models cross-modal similarity as evidence via Dirichlet distributions, employing a three-fold principle to identify query types and guide adaptive label calibration. Complementarily, at the intra-video level, we introduced flexible optimal transport with an adaptive dustbin to enable dense temporal alignment, effectively filtering out noise while accumulating reliable supervision signals, yielding superior performance.

## Limitations

For fair comparison, we follow prior works by using pretrained models (e.g., ResNet) for feature extraction, which may limit retrieval performance. In future work, we plan to adopt more advanced encoders (e.g., CLIP) and enable end-to-end training. Consistent with most retrieval methods, *Holmes* requires access to the full video, making it less suitable for online streaming video retrieval scenarios.

## Acknowledgements

We sincerely thank the reviewers and chairs for their efforts and constructive suggestions, which have helped us improve the manuscript. This work is supported in part by the National Natural Science Foundation of China under grants 624B2088, 62571298, 62536003, 62521006, and in part by the project of Peng Cheng Laboratory (PCL2025A14).

## Impact Statement

This paper introduces *Holmes*, a framework for PRVR that explicitly models semantic uncertainty and query ambiguity to enhance the reliability of AI systems. Beyond improving retrieval accuracy, a primary societal benefit of our work is the shift from deterministic "black-box" predictions to interpretable, uncertainty-aware retrieval. By distinguishing between confident matches and low-evidence guesses, our approach fosters trust in downstream applications and allows for human-in-the-loop verification. While advanced retrieval technologies carry potential dual-use risks, such as in automated surveillance, *Holmes* actively mitigates the harm of "false positives" through its intra-video evidential learning and adaptive dustbin mechanism, which filter out spurious correlations and noise. Furthermore, by explicitly categorizing polysemous and under-determined queries, our framework offers a methodological pathway to diagnose and analyze dataset biases rather than blindly overfitting to them. We believe that by prioritizing uncertainty quantification and the suppression of unreliable signals, this work steers the field toward more robust, transparent, and responsible multimodal video understanding.

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

# A. More Experiments

## A.1. Hyper-Parameters

We adopt the hyperparameter settings from HLFormer (Li et al., 2025b) and GMMFormerV2 (Wang et al., 2024a). The numbers of clips and frames are set to $M_c = 32$ and $M_f = 128$, respectively. The maximum query length $N_q$ is fixed to 64 for ActivityNet Captions and 30 for TVR and Charades-STA. All remaining settings, including $L_{sim}$ and $L_{div}$, follow (Li et al., 2025b). Detailed implementation can be provided in the submitted code ICML26-Holmes.

## A.2. Parameter Analysis

We performed a comprehensive hyperparameter analysis on the TVR dataset to investigate the effects of the temperature $\tau$ in Equation (3), the threshold $\beta$ in Equation (11), and the dustbin bucket ratio $z$ in Equation (20), as depicted in Figure 6. The results indicate that our model maintains relatively stable performance when $\tau$ is within [0.06, 0.2], $\beta$ within [0.2, 0.5], and $z$ within [0.2, 0.4], demonstrating that the proposed approach is largely insensitive to hyperparameter variations and exhibits strong robustness.

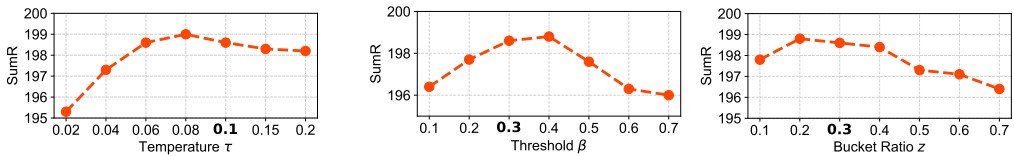

*Figure 6.* The parameter analysis of the temperature $\tau$ in Equation (3), the threshold $\beta$ in Equation (11), and the dustbin bucket ratio $z$ in Equation (20) on the TVR dataset, with the default settings highlighted in bold.

# B. More Details on Method

## B.1. Basic Training Objectives

Following existing works(Dong et al., 2022a; Wang et al., 2024b), we adopt triplet loss (Dong et al., 2021; Faghri et al., 2017) $L^{trip}$ and InfoNCE loss (Miech et al., 2020; Zhang et al., 2021) $L^{nce}$, query diverse loss (Wang et al., 2024b) $L_{div}$ that are widely used in PRVR. A text-video pair is considered positive if the video contains a moment relevant to the text; otherwise, it is regarded as negative. Given a positive text-video pair $(T, V)$, the triplet ranking loss over the mini-batch $\mathcal{B}$ is formulated as:

$$L^{trip} = \frac{1}{n} \sum_{(T,V) \in B} \{\max(0, m + s(T^-, V) - s(T, V)) + \max(0, m + s(T, V^-) - s(T, V))\},$$

where $m$ is a margin constant. $T^-$ and $V^-$ indicate a negative text for $V$ and a negative video for $T$, respectively. The similarity score $s(,)$ is obtained by Equation (1) and Equation (2).

The infoNCE loss is computed as:

$$L^{nce} = -\frac{1}{n} \sum_{(T,V) \in \mathcal{B}} \left\{\log\left(\frac{s(T,V)}{s(T,V) + \sum_{T_i^- \in \mathcal{N}_T} s(T_i^-, V)}\right) + \log\left(\frac{s(T,V)}{s(T,V) + \sum_{V_i^- \in \mathcal{N}_V} s(T, V_i^-)}\right)\right\},$$

where $\mathcal{N}_T$ and $\mathcal{N}_V$ represent the negative texts and videos of $V$ and $T$ within the mini-batch $\mathcal{B}$, respectively.

Finally , $L_{sim}$ is defined as:

$$L_{sim} = L_c^{trip} + L_f^{trip} + \lambda_c L_c^{nce} + \lambda_f L_f^{nce}, \tag{22}$$

where $f$ and $c$ mark the objectives for the frame-level branch and the clip-level branch, respectively. $\lambda_c$ and $\lambda_f$ are hyper-parameters to balance the contributions of InfoNCE objectives.

Given a collection of text queries $T$ in the mini-batch $\mathcal{B}$, the query diverse loss is defined as:

$$L_{div} = \frac{1}{n} \sum_{\boldsymbol{q}_i, \boldsymbol{q}_j \in T} \mathbb{1}_{\boldsymbol{q}_i, \boldsymbol{q}_j} \log\left(1 + e^{\alpha(cos(\boldsymbol{q}_i, \boldsymbol{q}_j) + \delta)}\right), \tag{23}$$

where $\delta > 0$ denotes the margin, $\alpha > 0$ is a scaling factor, and $\mathbb{1}_{\boldsymbol{q}_i, \boldsymbol{q}_j} \in \{0, 1\}$ represents an indicator function, $\mathbb{1}_{\boldsymbol{q}_i, \boldsymbol{q}_j} = 1$ when $\boldsymbol{q}_i$ and $\boldsymbol{q}_j$ correspond to the same video.

**B.2. Detailed proofs for Theorem 3.2**

*Proof.* We prove the theorem by providing a counter-example. Let $y_i$ be the ground-truth label corresponding to index $t$ (i.e., $y_{it} = 1$). Let $e_i$ be an evidence vector constructed such that all evidence is assigned to an incorrect video $k$ ($k \neq t$):

$$e_{ij} = \begin{cases} \lambda, & \text{if } j = k, \\ 0, & \text{otherwise,} \end{cases} \tag{24}$$

where $\lambda \gg 0$ is a large positive constant. According to Eq. (4) and Eq. (5), the Dirichlet strength is $S_i = \lambda + K$. Consequently, the epistemic uncertainty is $u_i = \dfrac{K}{\lambda + K}$. By choosing a sufficiently large $\lambda$, we can make $u_i$ arbitrarily close to 0 (low uncertainty).

However, the belief mass $b_{ij}$ is maximized at index $k$ because $\alpha_{ik} > \alpha_{ij}$ for all $j \neq k$. Since $k \neq t$, the highest belief is not assigned to the annotated label. This implies that a low epistemic uncertainty indicates that the model has collected a large amount of evidence, but it does not guarantee that this evidence points to the correct video. $\square$

**B.3. The derivation of aleatoric uncertainty in Equation (9)**

Following standard formulations in evidential learning (Li et al., 2025a; Chen et al., 2025a), let $\mathbf{p} = [p_1, \ldots, p_K]^\top$ be the probability vector for the $i$-th sample, defined on the $K$-dimensional unit simplex $\mathcal{S}^K$. We assume $\mathbf{p}$ follows a Dirichlet distribution parameterized by the concentration vector $\boldsymbol{\alpha}_i = [\alpha_{i1}, \ldots, \alpha_{iK}]^\top$. The probability density function (PDF) is defined as:

$$\text{Dir}(\mathbf{p}; \boldsymbol{\alpha}_i) = \begin{cases} \dfrac{1}{B(\boldsymbol{\alpha}_i)} \displaystyle\prod_{k=1}^{K} p_k^{\alpha_{ik}-1}, & \text{for } \mathbf{p} \in \mathcal{S}^K, \\ 0, & \text{otherwise,} \end{cases} \tag{25}$$

where $B(\boldsymbol{\alpha}_i) = \dfrac{\prod_{k=1}^{K} \Gamma(\alpha_{ik})}{\Gamma(S_i)}$ is the multivariate beta function, and $S_i = \sum_{k=1}^{K} \alpha_{ik}$ is the strength of the distribution. The expected entropy (aleatoric uncertainty) of the random variable $\mathbf{p}$ under this distribution is derived as follows:

$$\begin{aligned}
\mathbb{E}_{\mathbf{p} \sim \text{Dir}(\boldsymbol{\alpha}_i)}[H(\mathbf{p})] &= \int_{\mathcal{S}^K} \text{Dir}(\mathbf{p}; \boldsymbol{\alpha}_i) \left( -\sum_{k=1}^{K} p_k \ln p_k \right) d\mathbf{p} \\
&= -\sum_{k=1}^{K} \int_{\mathcal{S}^K} \text{Dir}(\mathbf{p}; \boldsymbol{\alpha}_i)(p_k \ln p_k) d\mathbf{p} \\
&= -\sum_{k=1}^{K} \int_{\mathcal{S}^K} \frac{\Gamma(S_i)}{\prod_{j=1}^{K} \Gamma(\alpha_{ij})} \prod_{j=1}^{K} p_j^{\alpha_{ij}-1}(p_k \ln p_k) d\mathbf{p} \\
&= -\sum_{k=1}^{K} \int_{\mathcal{S}^K} \frac{\alpha_{ik}}{S_i} \frac{\Gamma(S_i+1)}{\Gamma(\alpha_{ik}+1) \prod_{j \neq k} \Gamma(\alpha_{ij})} p_k^{\alpha_{ik}} \prod_{j \neq k} p_j^{\alpha_{ij}-1} \ln p_k d\mathbf{p} \\
&= -\sum_{k=1}^{K} \frac{\alpha_{ik}}{S_i} \int_{\mathcal{S}^K} \text{Dir}(\mathbf{p}; \boldsymbol{\alpha}_i + \mathbf{1}_k) \ln p_k d\mathbf{p} \\
&= -\sum_{k=1}^{K} \frac{\alpha_{ik}}{S_i} \mathbb{E}_{\mathbf{p} \sim \text{Dir}(\boldsymbol{\alpha}_i + \mathbf{1}_k)}[\ln p_k] \\
&= -\sum_{k=1}^{K} \frac{\alpha_{ik}}{S_i} (\psi(\alpha_{ik}+1) - \psi(S_i+1)) \\
&= \sum_{k=1}^{K} \frac{\alpha_{ik}}{S_i} (\psi(S_i+1) - \psi(\alpha_{ik}+1)),
\end{aligned}$$

where the term $\mathbf{1}_k$ represents a one-hot vector with the $k$-th element set to 1 and all others set to 0. The derivation primarily relies on the recursive property of the Gamma function, $\Gamma(z+1) = z\Gamma(z)$. This property allows us to rewrite the probability

density function by extracting the factor $\frac{\alpha_{ik}}{S_i}$, thereby transforming the original integral into an expectation over a new Dirichlet distribution with parameters shifted by $\mathbf{1}_k$ (i.e., $\boldsymbol{\alpha}_i + \mathbf{1}_k$).

### B.4. The derivation of loss function in Equation (17)

In this section, we provide the detailed derivation of the evidential learning objective presented in the main paper.

Our method aims to guide the predicted probability $\boldsymbol{p}_i$ to align with the calibrated label $\hat{\boldsymbol{y}}_i$. Since $\boldsymbol{p}_i$ is a random variable, we formulate the uncertainty-aware learning objective as the expected sum of squared errors between $\hat{\boldsymbol{y}}_i$ and $\boldsymbol{p}_i$. The derivation is as follows:

$$
\begin{aligned}
L_U(\boldsymbol{\alpha}_i, \hat{\boldsymbol{y}}_i) &= \int \|\hat{\boldsymbol{y}}_i - \boldsymbol{p}_i\|_2^2 \frac{1}{B(\boldsymbol{\alpha}_i)} \prod_{j=1}^K p_{ij}^{\alpha_{ij}-1} d\boldsymbol{p}_i \\
&= \int \|\hat{\boldsymbol{y}}_i - \boldsymbol{p}_i\|_2^2 \, \mathrm{Dir}(\boldsymbol{p}_i; \boldsymbol{\alpha}_i) \, d\boldsymbol{p}_i \\
&= \sum_{j=1}^K \mathbb{E}_{\boldsymbol{p}_i \sim \mathrm{Dir}(\boldsymbol{\alpha}_i)} \left[ (\hat{y}_{ij} - p_{ij})^2 \right] \\
&= \sum_{j=1}^K \left[ (\hat{y}_{ij} - \mathbb{E}[p_{ij}])^2 + \mathrm{Var}(p_{ij}) \right] \\
&= \sum_{j=1}^K \left( \hat{y}_{ij} - \frac{\alpha_{ij}}{S_i} \right)^2 + \frac{\alpha_{ij}(S_i - \alpha_{ij})}{S_i^2(S_i + 1)},
\end{aligned}
\tag{26}
$$

where $S_i = \sum_{j=1}^K \alpha_{ij}$. The third step utilizes the standard bias-variance decomposition, $\mathbb{E}[(X - c)^2] = (c - \mathbb{E}[X])^2 + \mathrm{Var}(X)$, and the final step substitutes the mean and variance of the Dirichlet distribution: $\mathbb{E}[p_{ij}] = \alpha_{ij}/S_i$ and $\mathrm{Var}(p_{ij}) = \alpha_{ij}(S_i - \alpha_{ij})/[S_i^2(S_i + 1)]$.

### B.5. Derivation of the Sinkhorn Algorithm

In this section, we provide a detailed derivation of the Sinkhorn algorithm used for calculating the optimal transport (Equation (19)). Consider a similarity matrix $\mathbf{S} \in \mathbb{R}^{n \times m}$, where $[\mathbf{S}]_{i,j}$ represents the similarity score between a video clip $\mathbf{c}_i$ and a query $\mathbf{q}_j$. The goal of optimal transport is to find a transport plan $\mathbf{Q}$ that maximizes the total similarity under marginal constraints. The primal problem is defined as:

$$
\begin{aligned}
\max_{\mathbf{Q} \in \mathcal{Q}} \quad & \langle \mathbf{Q}, \mathbf{S} \rangle = \mathrm{tr}(\mathbf{Q}^\top \mathbf{S}) = \sum_{i=1}^n \sum_{j=1}^m [\mathbf{Q}]_{i,j} \cdot [\mathbf{S}]_{i,j} \\
\text{s.t.} \quad & \mathcal{Q} = \left\{ \mathbf{Q} \in \mathbb{R}_+^{n \times m} \mid \mathbf{Q}\mathbf{1}_m = \boldsymbol{\mu}, \ \mathbf{Q}^\top \mathbf{1}_n = \boldsymbol{\nu} \right\},
\end{aligned}
\tag{27}
$$

where $\boldsymbol{\mu} \in \mathbb{R}^n$ and $\boldsymbol{\nu} \in \mathbb{R}^m$ are probability vectors representing the mass distribution of video clips and queries, respectively. For uniform distributions, we set $\boldsymbol{\mu} = \frac{1}{n}\mathbf{1}_n$ and $\boldsymbol{\nu} = \frac{1}{m}\mathbf{1}_m$.

Standard linear programming solvers for Eq. (27) typically require $O(n^3 \log n)$ time complexity, which is computationally prohibitive for large-scale datasets. To address this, Cuturi (Cuturi, 2013) proposed an entropy-regularized formulation that admits a fast approximate solution:

$$
\max_{\mathbf{Q} \in \mathcal{Q}} \quad \langle \mathbf{Q}, \mathbf{S} \rangle + \varepsilon H(\mathbf{Q}),
\tag{28}
$$

where $H(\mathbf{Q}) = -\sum_{i,j} [\mathbf{Q}]_{i,j} \ln([\mathbf{Q}]_{i,j})$ is the entropy regularization term and $\varepsilon > 0$ is the regularization coefficient. This convex objective function allows us to utilize the Lagrange multiplier method. The Lagrangian $\mathcal{L}(\mathbf{Q}, \boldsymbol{u}, \boldsymbol{v})$ with dual multipliers $\boldsymbol{u} \in \mathbb{R}^n$ and $\boldsymbol{v} \in \mathbb{R}^m$ is given by:

$$
\mathcal{L}(\mathbf{Q}, \boldsymbol{u}, \boldsymbol{v}) = \langle \mathbf{Q}, \mathbf{S} \rangle + \varepsilon H(\mathbf{Q}) + \boldsymbol{u}^\top (\mathbf{Q}\mathbf{1}_m - \boldsymbol{\mu}) + \boldsymbol{v}^\top (\mathbf{Q}^\top \mathbf{1}_n - \boldsymbol{\nu}).
\tag{29}
$$

Taking the partial derivative of Eq. (29) with respect to each entry $[\mathbf{Q}]_{i,j}$ and setting it to zero (KKT condition):

$$
\frac{\partial \mathcal{L}}{\partial [\mathbf{Q}]_{i,j}} = [\mathbf{S}]_{i,j} - \varepsilon(\ln([\mathbf{Q}]_{i,j}) + 1) + u_i + v_j = 0.
\tag{30}
$$

Solving for $[\mathbf{Q}]_{i,j}$, we obtain:

$$[\mathbf{Q}]_{i,j} = \exp\left(\frac{[\mathbf{S}]_{i,j} + u_i + v_j}{\varepsilon} - 1\right) = e^{-\frac{1}{2} + \frac{u_i}{\varepsilon}} \cdot e^{\frac{[\mathbf{S}]_{i,j}}{\varepsilon}} \cdot e^{-\frac{1}{2} + \frac{v_j}{\varepsilon}}. \tag{31}$$

This factorization suggests that the optimal solution $\mathbf{Q}^*$ can be expressed in terms of two scaling vectors, $\boldsymbol{\kappa}_1 \in \mathbb{R}^n$ and $\boldsymbol{\kappa}_2 \in \mathbb{R}^m$. By defining $[\boldsymbol{\kappa}_1]_i = \exp\left(-1/2 + u_i/\varepsilon\right)$ and $[\boldsymbol{\kappa}_2]_j = \exp\left(-1/2 + v_j/\varepsilon\right)$, the optimal transport matrix takes the form of a normalized exponential matrix:

$$\mathbf{Q}^* = \mathrm{Diag}(\boldsymbol{\kappa}_1) \exp(\mathbf{S}/\varepsilon) \mathrm{Diag}(\boldsymbol{\kappa}_2). \tag{32}$$

Since $\mathbf{Q}^*$ must satisfy the marginal constraints in Eq. (27), we have:

$$\mathbf{Q}^* \mathbf{1}_m = \boldsymbol{\mu} \quad \text{and} \quad \mathbf{Q}^{*\top} \mathbf{1}_n = \boldsymbol{\nu}. \tag{33}$$

Substituting Eq. (32) into these constraints leads to the Sinkhorn-Knopp fixed-point iteration updates:

$$\boldsymbol{\kappa}_1 \leftarrow \boldsymbol{\mu} \oslash (\exp(\mathbf{S}/\varepsilon)\boldsymbol{\kappa}_2), \quad \boldsymbol{\kappa}_2 \leftarrow \boldsymbol{\nu} \oslash (\exp(\mathbf{S}^\top/\varepsilon)\boldsymbol{\kappa}_1), \tag{34}$$

where $\oslash$ denotes element-wise division. Empirically, a few iterations (e.g., 50 steps) are sufficient to converge to a satisfactory transport plan $\mathbf{Q}^*$. The final optimal transport distance is then computed as $\langle \mathbf{Q}^*, \mathbf{S} \rangle$.

### B.6. Further details of Our Architecture

We replace the standard self-attention in the Transformer block with Gaussian attention (Wang et al., 2024b), yielding a Gaussian block. Multiple Gaussian blocks ($N_g$) operate in parallel, and their outputs are aggregated via MAIM (Li et al., 2025b) to construct the frame-/clip-level encoder. For notational convenience, we unify the frame and clip embeddings as $\boldsymbol{V}_o$, since they are processed in an identical manner. Given $\boldsymbol{V}_o$, the Gaussian attention is defined as:

$$\mathrm{GA}(\boldsymbol{V}_o) = \mathrm{softmax}\left(\mathcal{M}_\sigma^g \odot \frac{\boldsymbol{V}_o W^q (\boldsymbol{V}_o W^k)^\top}{\sqrt{d_h}}\right) \boldsymbol{V}_o W^v, \tag{35}$$

where $\mathcal{M}_\sigma^g(i,j) = \frac{1}{2\pi} e^{-\frac{(j-i)^2}{\sigma^2}}$ is a Gaussian kernel with variance $\sigma^2$. Varying $\sigma$ enables modeling feature interactions at different temporal scales, yielding multi-receptive-field representations. $W^q$, $W^k$ and $W^v$ denote linear projections. $d_h$ is the attention dimension and $\odot$ indicates element-wise multiplication.

### B.7. Further discussion of related work

Existing evidential learning methods have been applied to noisy cross-modal retrieval and text-to-video retrieval, In noisy settings, prior works(Qin et al., 2022; Zha et al., 2025) leverage epistemic uncertainty derived from Dirichlet parameters to identify instance-level noisy pairs (i.e., clean vs. noisy). In text-video retrieval, (Liu et al., 2025) integrates probabilistic embeddings with epistemic uncertainty only as a regularization signal. They do not explicitly model query-level semantic heterogeneity and also fail to address the unique challenges of PRVR.

Holmes introduces a hierarchical evidential framework specifically designed for PRVR:

- Inter-video: PRVR queries exhibit fine-grained heterogeneity beyond a simple clean/noisy dichotomy. We propose a three-fold principle that goes beyond epistemic uncertainty by incorporating label consistency and aleatoric uncertainty for query diagnosis, enabling cross-scale conflict-aware fusion and query-adaptive label calibration.

- Intra-video: PRVR relies on MIL to handle partial relevance, which induces sparse supervision. We employ FOT to provide denser supervision, generating reliable intra-video evidence to support inter-video learning.

