# OpenReview forum: "Revisiting Uncertainty: On Evidential Learning for Partially Relevant Video Retrieval"
_ICML.cc/2026/Conference — ICML 2026 regular_

### Official Review · Reviewer_S9zY · 2026-02-23

**Soundness:** 4
**Presentation:** 3
**Significance:** 4
**Originality:** 4
**Overall Recommendation:** 5
**Confidence:** 4

**Summary:**

This paper presents Holmes, a novel method for partially relevant video retrieval (PRVR), built on a theoretically grounded dual-level evidential learning framework that explicitly models uncertainty arising from data asymmetry.
At the inter-video level, cross-modal similarities are treated as evidence under Dirichlet distributions to estimate both epistemic and aleatoric uncertainty, and an innovative three-fold principle categorizes queries into precise, polysemous, and under-determined types with query-adaptive label calibration.
At the intra-video level, a flexible optimal transport formulation with a dustbin bucket yields soft, denser alignments, mitigating the sparse supervision inherent in MIL. Extensive experiments on three benchmarks demonstrate state-of-the-art performance, supported by comprehensive ablation and efficiency analyses.

**Compliance With Llm Reviewing Policy:**

Affirmed.

**Final Justification:**

The authors have address all my concern and I would like to keep my rating.

**Key Questions For Authors:**

Please refer to the weaknesses outlined above; I also have several questions below：

1. What is the sensitivity of results to the evidence mapping function (exp(tanh(s/τ)))? Have you tried other alternatives (e.g., softplus, ReLU)?
2. In evidential learning, a Kullback–Leibler (KL) divergence term is usually introduced to penalize deviations under uncertain retrieval. Why is such a regularization term not incorporated in Holmes?
3. Do the authors plan to release the code publicly？

My concerns mainly relate to the clarification of certain methodological details. I look forward to the authors’ responses to the identified weaknesses and questions, and I will carefully consider adjusting my score based on the rebuttal.

**Limitations:**

The authors provide an Impact Statement but do not explicitly discuss the limitations of the work. I encourage the authors to analyze potential failure cases.

**Strengths And Weaknesses:**

### Strengths
1. The core idea  for leveraging evidential learning to address inter-video semantic ambiguity and intra-video sparse supervision in PRVR is well-motivated. The authors further propose a theoretically rigorous three-fold principle to categorize query types, along with a flexible optimal transport formulation for intra-video alignment. Overall, Holmes demonstrates clear novelty and conceptual coherence.
2. The paper provides a detailed model analysis, supported by both qualitative and quantitative results. These analyses validate the advantages of the proposed evidential learning approach for PRVR.
3. Comparisons with prior methods show consistent gains across all three datasets and these improvements are further reinforced by the efficiency analysis.
4. Code is also available.

### Weaknesses
1. The Uncertainty Guided Identification  contains a sign error in  Eq. (12). According to Fig. 2(b) and Lines 199–204, precise queries should exhibit lower aleatoric uncertainty, and thus the condition should be “< median”.
2. There is a minor typos in Line 207: it may be Proposition 3.4 rather than Theorem 3.4.
3. Several implementation details of Holmes remain unclear. For instance, the mathematical formulation of $L_{intra}$ is not provided, and the transformation from fused beliefs $(b^o, u^o)$ back to Dirichlet parameters $\alpha^o$ for use in $L_U$ is not explicitly specified.
4. The paper does not explicitly discuss the limitations of Holmes.

---

> ### Author Rebuttal · Authors · 2026-03-27
>
> Dear Reviewer `S9zY`,
>
> We sincerely thank you for your review and valuable suggestions!  Happy to hear you enjoy our **well-motivated method**, **clear novelty**, **detailed analysis** and **available code**.  Below, we address each of your concerns in detail. *If you find them satisfactory, we would appreciate your consideration in updating your scores*.
>
> > **W1**&**W2**: Sign error in Eq.(12) and a minor typo in Line 207.
>
> **A1**: Many thanks. You are correct that it should be “< median” rather than “> median” in Eq. (12) and this has been corrected. We have also revised “Theorem 3.4” to “Proposition 3.4”.
>
> > **W3**: Clarification: the formulation of $L_{intra}$ and the transformation from $b^o,u^o$ back to $\alpha^o$.
>
> **A2**: Sorry for any confusion.
>
> For the formulation of $L_{\text{intra}}$: we model intra-video alignment as an $M_c$-way classification problem, similar to inter-video evidential learning formulated as a $K$-way cross-modal classification task. Let $\mathbf{S} \in \mathbb{R}^{M_c \times M_q}$ denote the clip–query similarity matrix of a video, where $M_c$ and $M_q$ represent the numbers of clips and queries corresponding to the video, respectively. Based on Eq.(3) and Eq.(4), we can collect evidence to estimate the intra-video Dirichlet parameters $\alpha_{\text{intra}}$. We then treat $\bar{\mathbf{Q}}^*_{1:M_c,1:M_q}$ as pseudo-labels, and obtain query-wise intra-video labels via transposition and normalization:
> $Q_{\text{intra}} = \mathrm{Norm}(\bar{\mathbf{Q}}^{*T}).$ The final objective is defined as: $L_{\text{intra}} = L_U(\alpha_{\text{intra}}, Q_{\text{intra}})$.
>
>
> For the transformation from $b^{o}$, $u^o$ to $\alpha^o$: based on Eq.(5), we obtain:
> $ S^{o} = \frac{K}{u^{o}}$, ${\alpha_j}^{o}$ = $1+S^{o} {b_j}^{o}$, $\alpha^o=[\alpha^o_{1},\alpha^o_{2},\dots,\alpha^o_{K}]$.
> > **W4**: Need to discuss Holmes's limitations.
>
> **A3**: We thank the reviewer for highlighting this.  We have added a discussion in the Conclusion to explicitly address the main limitations of Holmes: 1. For fair comparison, we follow prior works by using pre-trained models (e.g., ResNet and RoBERTa) for feature extraction, which may limit retrieval performance. In future work, we plan to adopt more advanced encoders (e.g., CLIP) and enable end-to-end training; 2. Consistent with most retrieval methods, Holmes requires access to the full video, making it less suitable for online streaming video retrieval scenarios.
>
> > **Q1**:  The sensitivity of results to the evidence mapping function.
>
> **A4**: In **Appendix A.2**, we investigate the effects of the temperature $\tau$ in $\exp(\tanh(s/\tau))$. The results show that the model remains relatively stable for $\tau \in [0.06, 0.2]$. We also explore alternative functions (e.g., Softplus and ReLU) on the TVR dataset, with results summarized in the table below:
>
> |           | R@1  | R@5  | R@10 | R@100 | SumR  |
> |-----------|------|------|------|-------|-------|
> | exp(tanh) | 18.4 | 40.7 | 52.0 | 87.5  | 198.6 |
> | softplus  | 18.3 | 40.6 | 52.1 | 87.4  | 198.3 |
> | relu      | 18.2 | 40.6 | 52.0 | 87.4  | 198.2 |
>
> As shown, $\exp(\tanh(s/\tau))$ achieves better overall performance.
>
>
> > **Q2**: Why is the  KL divergence term not incorporated in Holmes?
>
> **A5**: Standard Evidential Learning methods often employ a KL divergence term to forcefully shrink the evidence of non-ground-truth classes to zero. However, in the context of PRVR, we frequently encounter polysemous queries. This means multiple video candidates may contain valid semantics. Applying a strict KL penalty would aggressively force the model to ignore these semantically related samples, thereby introducing misleading supervision. Therefore, we do not use the KL divergence term.
>
> > **Q3**:  Do the authors plan to release the code publicly？
>
> **A6**: Sure. We have provided the anonymous code for review. Upon acceptance, we will release the complete code, including all implementations, model weights, detailed training configurations and logs.

---

> > ### Author Rebuttal · Reviewer_S9zY · 2026-04-01
> >
> > The authors have address all my concern and I would like to keep my rating.

---

> > > ### Author Response · Authors · 2026-04-01
> > >
> > > We sincerely appreciate your recognition of our paper. We are truly encouraged to know that your concerns have been fully addressed and that you have  maintained your score of 5. We are deeply grateful for your time, thoughtful feedback, and continued support.

---

### Official Review · Reviewer_VJc5 · 2026-03-05

**Soundness:** 3
**Presentation:** 3
**Significance:** 2
**Originality:** 1
**Overall Recommendation:** 3
**Confidence:** 4

**Summary:**

This paper proposes Holmes, a hierarchical evidential learning framework designed for Partially Relevant Video Retrieval (PRVR). To address the inherent uncertainty caused by semantic ambiguity across videos and sparse supervision within videos, the authors leverage Evidential Deep Learning (EDL) to explicitly quantify uncertainty using Dirichlet distributions at the inter-video level. Furthermore, they employ Flexible Optimal Transport (FOT) with an adaptive dustbin mechanism to aggregate dense temporal evidence at the intra-video level.

**Compliance With Llm Reviewing Policy:**

Affirmed.

**Final Justification:**

Thank you for the detailed rebuttal and discussion. This paper proposes a hierarchical evidential learning framework for PRVR, which is an interesting direction.

However, the authors do not provide a clear methodological distinction from existing evidential learning approaches, making it difficult to assess the **true novelty beyond a task-specific adaptation**.

In addition, I remain concerned about the reliability of the learned evidence. The proposed strategies (e.g., warm-up, consistency, variance regularization) **do not fully address the issue that model-predicted evidence itself may be unreliable**, especially under confident but incorrect predictions.

Overall, while the paper is interesting, the concerns on novelty and reliability remain. I therefore keep my **Weak Reject** decision unchanged.

**Key Questions For Authors:**

1. In Table 1, the retrieval accuracy (e.g., R@1) for the **PRVR** task is consistently and significantly higher than that of the standard **T2VR** task across multiple datasets. According to the Introduction, PRVR is inherently more challenging due to the partial relevance and the massive amount of irrelevant background noise in untrimmed videos. Could the authors explain why the experimental results appear to contradict this task-difficulty intuition?
2. The strategies proposed for the two major challenges— Evidential Deep Learning for inter-video semantic ambiguity and Flexible Optimal Transport (FOT) for intra-video sparse supervision—seem **disjointed** in the current methodology. Could the authors clarify if there is a deeper intrinsic connection between these two components beyond being a "**combined pipeline**"? Specifically, would it be possible (or did the authors attempt) to extend the Evidential Uncertainty Modeling to the intra-video level to directly address sparse supervision, rather than relying on a separate FOT mechanism? Exploring this would significantly strengthen the unity and conceptual contribution of the framework.

**Limitations:**

yes

**Strengths And Weaknesses:**

***Strengths***

1. **Soundness**: The technical approach is generally sound and well-motivated. The inclusion of source code further enhances the reproducibility of the proposed method. Additionally, the theoretical components are supported by proofs, making the overall methodology rigorous and reasonable.
2. **Presentation**: The paper is well-written and organized. The illustrations and tables are clear, which facilitates a smooth understanding of the core concepts and experimental results.



***Weaknesses***

1. **Significance: Generality of the Conceptual Contribution**

   While the authors highlight their conceptual contributions in addressing **inter-video semantic ambiguity**, it should be noted that this challenge is **not unique** to the PRVR task. Semantic ambiguity is a fundamental issue across various retrieval tasks, such as general text-to-image or text-to-video retrieval, and is not exclusively triggered by the "partially relevant" nature of PRVR. Therefore, it is difficult to claim that the identification of this challenge constitutes a distinct conceptual contribution specific to the PRVR domain.

2. **Originality: Insufficient Discussion of Related Work**

   The current manuscript lacks a deep dive into existing literature, which hinders a thorough assessment of its originality:

   - **Uncertainty Learning:** There is a lack of in-depth discussion regarding existing uncertainty learning methods, particularly those based on Dirichlet distributions. The authors need to clearly articulate the specific innovations of their approach compared to prior evidential learning works.
   - **Optimal Transport (OT):** Regarding the intra-video sparse supervision issue, the paper seems to imply that the Optimal Transport mechanism is the default or only solution. The authors should clarify why existing methods fail to address this specific challenge and why their proposed FOT is uniquely required.

3. **Presentation: Figure 1 Improvement**

   Figure 1 effectively illustrates several classic examples of the challenges mentioned. However, there is currently no explicit mapping between these specific examples and the two core challenges. I suggest revising Figure 1 to explicitly label which examples correspond to Challenge 1 and which correspond to Challenge 2 for better clarity.

4. **Technical Soundness: Heuristic Assumption in Adaptive Dustbin**

   In the adaptive dustbin component of the FOT, the authors set a threshold at the **30% percentile**. This represents a very strong prior assumption, implying that approximately 30% of the segments in all videos are irrelevant noise. However, in PRVR, the ratio of relevant segments varies significantly across different videos and datasets. A fixed ratio may lead to "over-filtering" (relevant segments being absorbed into the dustbin) or "under-filtering" (noise remaining in the transport plan), potentially compromising the model's robustness across diverse video lengths.

---

> ### Author Rebuttal · Authors · 2026-03-30
>
> Dear Reviewer ` VJc5`,
>
> Thanks for the insightful comments. We sincerely appreciate the reviewer's positive comments on **well-motivated method**, **well-written paper** and **clear presentation**. Below are our point-by-point responses to your concerns. *If you find them satisfactory, we would appreciate your consideration in updating your scores*.
>
> > **W1** : Generality of the conceptual contribution.
>
> **A1**:
> - Semantic ambiguity, while present in general retrieval,  is **fundamentally exacerbated** in PRVR due to the severe data asymmetry between brief queries and rich, untrimmed video content. Unlike standard text-to-video retrieval with fully aligned descriptions, PRVR queries correspond only to partial segments, inevitably inducing **severe semantic ambiguity across multiple videos**. Moreover,  MIL in PRVR results in sparse supervision and insufficient intra-video evidence to resolve ambiguity, **further intensifying** semantic ambiguity.
> - Beyond identifying these issues, we quantify and disentangle its fine-grained heterogeneity via a tailored three-fold principle for this asymmetric partial retrieval setting. To mitigate MIL-induced sparsity, we integrate flexible optimal transport with evidential learning to produce denser supervision, yielding more reliable evidence for ambiguity identification. And  we believe these represent solid contributions.
>
> > **W2** : Insufficient discussion of related work.
>
> **A2**: Sorry for the confusion.
>
> -  Prior evidential learning methods primarily focus on epistemic uncertainty and remain largely unexplored in PRVR. Holmes introduces the first evidential learning framework for PRVR with two key innovations: 1. Inter-video learning, which augments epistemic uncertainty with label consistency and aleatoric uncertainty via a novel three-fold principle for fine-grained query identification with conflict-aware fusion and query-adaptive label calibration; 2. Intra-video learning, which integrates FOT with evidential learning to produce dense supervision, addressing intra-video sparsity overlooked in prior work.
> - Existing PRVR methods usually rely on  MIL, supervising only the best-matching clip, which induces  sparsity and susceptibility to spurious local noise. While OT can alleviate sparsity via dense alignments, its full source–target matching is ill-suited for PRVR, where queries correspond only to partial clips, leaving many redundant or noisy.  FOT introduces an adaptive dustbin to absorb noisy clips and mitigate misalignment, enabling dense temporal supervision.
>
> > **W3** : Figure 1 improvement.
>
> **A3**: Good idea. We will revise Figure 1 to align examples with their corresponding challenges.  Subfigures (a)–(d) will be grouped and labeled under “Challenge 1: Inter-video Semantic Ambiguity,” while subfigure (e) will be labeled as “Challenge 2: Intra-video Sparse Supervision".
>
> > **W4** : Heuristic assumption in adaptive dustbin.
>
> **A4**:  Standard optimal transport ignores redundant video clips, we thus introduce a dustbin bucket with the threshold $z$  to filter noisy clips. ID(0) and ID(11-12) in Table 3 show that this simple strategy yields gains. Meanwhile, the parameter analysis in **Appendix A.2** shows that Holmes is robust to $z$. On TVR dataset, performance remains stable when $z$ is within [0.2, 0.4] and results on Charades (see response to `Reviewer M8Q3's W2-1`) also verify this robustness.  $z$ is fixed across datasets without dataset-specific tuning; further tuning $z$ per dataset can do better.
>
> > **Q1** : Why does PRVR achieve higher retrieval accuracy than T2VR?
>
> **A5**: Following prior work, we retrain and evaluate T2VR methods on PRVR-specific datasets, rather than reporting the performance on T2VR tasks. T2VR methods are designed for pre-trimmed videos with fully relevant queries. When applied to PRVR tasks, their global representations are affected by  irrelevant background, resulting in mismatches with partial queries and degraded performance. PRVR models are  designed for partially relevant retrieval, naturally yielding better accuracy.
>
> > **Q2** : Clarify if there is an intrinsic connection between inter-video evidential learning and FOT .
>
> **A6**: We clarify that the two components are connected within a unified hierarchical evidential learning framework. We indeed have extended evidential learning to the intra-video level. As detailed in Section 3.3, FOT is not a disjoint module; instead, it produces soft one-to-many assignments ($\bar{Q}^*$), which serve as calibrated label for intra-video learning. We view intra-video alignment as a classification problem, where Dirichlet parameters are estimated from clip-level evidence, and the  evidential objective ($L_{intra}$) is used to promote evidence accumulation for relevant clips (see response to `Reviewer S9zY's W3` for $L_{intra}$). This design yields denser supervision and more reliable intra-video evidence, which enhances inter-video evidential learning, forming a coherent framework.

---

> > ### Author Rebuttal · Reviewer_VJc5 · 2026-04-04
> >
> > Thank you for your detailed rebuttal. I appreciate the clarifications, and several of my initial concerns have been addressed.
> >
> > However, some issues remain unresolved:
> >
> > 1. **Uncertainty Learning**: You introduce Dirichlet distributions for inter-video evidential learning, but fail to discuss whether similar methods exist in other domains or could be adapted to PRVR. Without this, it is difficult to assess the **novelty** of your approach.
> >
> > 2. **Inter-video vs. FOT**: My original question explicitly asked about the connection between **inter-video** evidential learning and FOT. Your response focuses almost entirely on **intra-video** learning, which does not answer the question.
> >
> > 3. **Evidence reliability**: The framework relies on model-predicted Dirichlet evidence, which is inherently **uncertain**. It is unclear how mis-predicted evidence is handled.

---

> > > ### Author Response · Authors · 2026-04-04
> > >
> > > ### Thank you for your detailed follow-up comments. We are happy to hear that our rebuttal has addressed some concerns. We are especially grateful that you have offered us the opportunity to provide further clarification for your specific remaining concerns. If any part of our response remains unclear, please do not hesitate to point it out, either in the acknowledgment block or in the original review block. Should our responses address your concerns, we would greatly appreciate your reconsideration of the score. Thank you very much in advance!
> > >
> > > > **FQ1**: Whether similar methods exist in other domains or could be adapted to PRVR.
> > >
> > > **FA1**:  Existing evidential learning methods have been applied to noisy cross‑modal retrieval and text‑to‑video retrieval, In noisy settings, prior works [1,2] leverage epistemic uncertainty derived from Dirichlet parameters to identify instance-level noisy pairs (i.e., clean vs. noisy). In text-video retrieval, [3] integrates probabilistic embeddings with epistemic uncertainty only as a regularization signal. They do not explicitly model query-level semantic heterogeneity and also fail to address the unique challenges of PRVR.
> > >
> > > Holmes introduces a hierarchical evidential framework specifically designed for PRVR:
> > >
> > > - Inter-video: PRVR queries exhibit fine-grained heterogeneity beyond a simple clean/noisy dichotomy. We propose a three-fold principle that goes beyond epistemic uncertainty by incorporating label consistency and aleatoric uncertainty for query diagnosis,  enabling cross-scale conflict-aware fusion and query-adaptive label calibration.
> > > - Intra-video:  PRVR relies on MIL to handle partial relevance, which induces sparse supervision and sensitivity to local noise. We employ FOT to provide denser supervision via $\bar{\mathbf{Q}}^*$, generating reliable intra‑video evidence that supports inter‑video learning.
> > >
> > > We hope these clarifications help reassess the novelty of Holmes, and we kindly request your reconsideration of our work.
> > >
> > > ---
> > > **References**
> > >
> > > [1] Deep Evidential Learning with Noisy Correspondence for Cross-modal Retrieval
> > >
> > > [2] UCPM: Uncertainty-Guided Cross-Modal Retrieval With Partially Mismatched Pairs.
> > >
> > > [3] DUQ: Dual Uncertainty Quantification for Text-Video Retrieval
> > >
> > > > **FQ2**: Inter-video vs. FOT.
> > >
> > > **FA2**: The relationship between inter‑video learning and FOT is as follows:
> > >
> > > - **The problem**: MIL in PRVR provides sparse intra‑video supervision, which leads to insufficient evidence for inter‑video discrimination and amplifies semantic ambiguity.
> > > - **The role of FOT**: FOT introduces soft one‑to‑many assignments, generating denser supervision for $L_{intra}$. This yields more robust and discriminative clip‑level embeddings.
> > > - **Impact on inter‑video evidence**: The improved embeddings enhance clip‑level inter‑video similarities ($s^c$), where inter‑video evidence ($e^c = \exp(\tanh(s^c/\tau))$) is derived. This increases the reliability of inter-video Dirichlet parameters.
> > > - **General Takeaways**: Without FOT, noisy clip representations would impair inter‑video evidence and uncertainty learning. FOT is therefore essential for enabling reliable inter‑video evidence under MIL.
> > >
> > >
> > > > **FQ3**: Evidence reliability: It is unclear how mis-predicted evidence is handled.
> > >
> > > **FA3**:  To clarify, **Holmes incorporates three complementary mechanisms** to ensure evidence reliability:
> > >
> > > 1. **Two‑Stage Warm‑up Strategy** (Lines 316–320): The model does not trust its predicted evidence from the beginning. A warm‑up stage uses standard losses ($L_{sim}$, $L_{div}$) and basic evidential learning without query label recalibration, allowing the model to first establish a reliable semantic baseline.
> > >
> > > 2. **Three‑fold Principle for Cross‑Validation**: Instead of relying solely on epistemic uncertainty, Holmes jointly considers label consistency and aleutic uncertainty. A confident but incorrect prediction would be flagged by the label consistency check, filtering out anomalous evidence.
> > >
> > > 3. **Variance Penalty in the Loss** (Eq.17 & Appendix B.4): The evidential loss includes a variance term $\frac{\alpha_{ij}(S_i-\alpha_{ij})}{S_i^2(S_i+1)}$ that inherently penalizes highly uncertain or conflicting evidence distributions, encouraging more robust estimation.
> > >
> > > We admit that the description of these mechanisms could be more centralized and prominent. We will revise the manuscript to explicitly present them as a coherent strategy for handling mis‑predicted evidence. Thank you for your reminder!

---

### Official Review · Reviewer_sJ65 · 2026-03-10

**Soundness:** 2
**Presentation:** 2
**Significance:** 3
**Originality:** 3
**Overall Recommendation:** 4
**Confidence:** 3

**Summary:**

The paper proposes \textit{Holmes}, a hierarchical evidential learning framework for Partially Relevant Video Retrieval (PRVR). The core motivation is to address the inherent uncertainty and supervisory sparsity in PRVR. At the inter-video level, it uses Dempster-Shafer Theory (via Dirichlet distributions) to model cross-modal similarity as evidence, categorizing queries into precise, polysemous, and under-determined sets to guide adaptive label calibration. At the intra-video level, it employs Flexible Optimal Transport (FOT) with a ``dustbin'' mechanism to filter out noisy, irrelevant clips and provide dense temporal alignment. Extensive experiments on ActivityNet Captions, Charades-STA, and TVR demonstrate state-of-the-art performance.

**Compliance With Llm Reviewing Policy:**

Affirmed.

**Final Justification:**

My concerns have been addressed, I would like raise my score to 4.

**Key Questions For Authors:**

1. **OT Normalization:** Please explain how the Dirichlet loss remains mathematically valid when applied to the non-normalized transport plan $\bar{Q}^*$ after the dustbin column is removed. Does this cause gradient instability?

2. **Median Split Justification:** How does the model perform if the true distribution of precise vs. polysemous queries is heavily skewed (e.g., 90/10)? Please provide an ablation or sensitivity analysis replacing the median split with a tunable percentile threshold.

3. **Fusion Strategy:** Table 3 shows UDPF outperforms Confidence Dominance. However, why use a hard rule at all? Why not use a continuous, uncertainty-weighted fusion of the frame and clip branches?

4. **Uncertainty in Evaluation:** Why does the evaluation phase not leverage the uncertainty-based results? Can an analysis be appended to show if uncertainty scores correlate with retrieval failure, or if they can be used to reject unanswerable queries during inference?

**Limitations:**

Maybe add a limitations discussion at the conclusion.

**Strengths And Weaknesses:**

**Strengths**

* **Comprehensive Experimental Validation:** The authors evaluate *Holmes* across three standard benchmarks (ActivityNet, Charades-STA, TVR) and compare it against a wide array of baselines (T2VR, VCMR, and PRVR). The proposed method consistently achieves state-of-the-art results (Table 1).
* **Thorough Ablation Studies:** Table 3 provides a very detailed breakdown of the framework's components. The authors successfully isolate the impact of multi-scale learning, fusion strategies, label calibration, intra-video learning, and the FOT mechanism, proving that each module contributes positively to the final performance.
* **Efficiency Maintained:** Table 2 is a strong addition. It demonstrates that *Holmes* achieves superior performance without introducing significant computational overhead during training or inference compared to similar-scale models like GMMFormerV2.

**Weaknesses**

**A. Methodological & Theoretical Flaws**
* **Arbitrary Heuristics in Query Partitioning:** Splitting queries using the median of aleatoric uncertainty forces exactly a 50/50 split between "polysemous" and "precise" queries within that subset. This is a massive inductive bias that assumes the dataset's true distribution of query types is perfectly balanced, which is highly unlikely in real-world scenarios.
* **Mathematical Inconsistency in Optimal Transport:** The FOT mechanism introduces a dustbin column, which is then dropped to obtain \(\bar{Q}^*\). Because the missing mass goes to the dustbin, the rows of \(\bar{Q}^*\) no longer sum to 1. Applying a Dirichlet classification loss (designed for valid probability distributions) to a truncated, non-normalized matrix is mathematically unsound.

**B. Experimental & Presentation Weaknesses**
* **Lack of Uncertainty Utilization in Evaluation:** The proposed method heavily relies on uncertainty modeling during training, yet the evaluation phase does not seem to leverage these uncertainty-based results. It is unclear why the estimated uncertainty is discarded at inference time.
* **Notation Errors:** There is a notational mistake in Definition 3.3 where the scalar \(s\) and the vector \(\mathbf{s}\) are misused, leading to mathematical ambiguity.
* **Lack of Sensitivity Analysis on Hyperparameters:** The method relies heavily on thresholds and hyperparameters (e.g., the temperature \(\tau\), the median split, the dustbin capacity in FOT). The experiments lack a sensitivity analysis showing how robust the model is to changes in these heavily engineered heuristics.

---

> ### Author Rebuttal · Authors · 2026-03-29
>
> Dear Reviewer ` sJ65`,
>
> We sincerely thank you for your constructive suggestions. We appreciate your recognition of Holmes’ **comprehensive and thorough validation**, as well as its **maintained efficiency**. Below, we address your concerns in a point-by-point manner. *If you find them satisfactory, we would appreciate your consideration in updating your scores*.
>
> > **W-A-1 & Q2**:  Arbitrary heuristics in query partitioning and median split justification.
>
> **A1**:  An insightful suggestion!
>
> - Holmes is built upon three principles for fine-grained query categorization. While epistemic uncertainty and label consistency enable an initial separation, they are insufficient to distinguish latent polysemous queries within the set of initially identified precise queries. Therefore, we further incorporate aleatoric uncertainty and adopt a simple median split. ID(0) and ID(9) in Table 3 show that such a simple partition already improves retrieval performance, which is  supported by Figure 5.
>
> - We also agree that using a tunable percentile threshold $\rho$ provides a more flexible and robust alternative. As suggested, we test  $\rho$ under a highly skewed distribution of precise vs. polysemous queries (e.g., 90/10), where $\rho=0$ denotes no separation and $\rho=100\%$ assigns all queries as polysemous.
>
> |SumR|77.4|78.8|78.6|78.5|78.0|77.5|
> |---------|----|----|----|----|----|----|
> |$\rho$(%)|0|10|30|50|75|100|
>
> As shown, a tunable $\rho$ improves performance and remains robust within  [10\%, 50\%]. Compared to no separation ($\rho$=0),  the median split ($\rho$=50%) yields  gains. We will add this analysis in the appendix.
>
> > **W-A-2 & Q1**: Missing OT normalization.
>
> **A2**: We respectfully clarify that we have applied row-wise normalization to $\bar{\mathbf{Q}}^*$ in the implementation. Please refer to Lines 111–114 in `src/Losses/fot.py` of the submitted code:
>
> ```python
>         align_logit_exp = align_logit_exp * text_mask.unsqueeze(-1)
>         row_sums = align_logit_exp.sum(dim=2, keepdim=True) + 1e-8
>         align_logit_exp = align_logit_exp / row_sums #row normalization
> ```
> `align_logit_exp` corresponds to $\bar{\mathbf{Q}}^*$. Therefore, no gradient instability is caused after normalization.
>
> > **W-B-1 & Q4**:  Why not use uncertainty in evaluation?
>
> **A3**:  For fair comparison, we follow prior work by employing only standard feature extraction and similarity computation at inference time. So uncertainty scores are not used to enhance retrieval at inference time. Notably, uncertainty scores correlate with retrieval failure: high epistemic uncertainty indicates poor query understanding and often leads to general retrieval failure, while high aleatoric uncertainty reflects inherent semantic ambiguity, making precise retrieval challenging. These signals can guide decisions such as multi-turn retrieval or query rewriting.  Uncertainty scores are useful for handling unanswerable queries (e.g., out-of-domain cases). Please refer to the response to `Reviewer M8Q3’s Q2`.
>
> > **W-B-2**:  Notation errors.
>
> **A4**: Sorry for the confusion. We have revised Eq. (7) in Definition 3.3 as: $c_i = \max(0, \mathbf{s}_i \cdot \mathbf{y}_i)$, where $\mathbf{s}_i$ denotes the similarity scores and $\mathbf{y}_i$ is the one-hot ground-truth label.
>
> > **W-B-3**: Lack of sensitivity analysis on hyperparameters.
>
> **A5**: In our response to `W-A-1 & Q2`, we present a sensitivity analysis of both the median split and the tunable threshold $\rho$.  In **Appendix A.2**, we conduct a comprehensive hyperparameter study on the TVR dataset, including temperature $\tau$ and the dustbin bucket ratio $z$ in FOT. We also provide  results on the Charades dataset (see response to `Reviewer M8Q3’s W2-1`). Overall, the results demonstrate that Holmes maintains stable performance within a reasonable hyperparameter range. Notably, all hyperparameters are fixed across datasets without dataset-specific tuning; further tuning per dataset can  enhance performance.
>
> > **Q3**: Why not use a continuous, uncertainty-weighted fusion?
>
> **A6**:  Holmes incorporates both strategies. For continuous aggregation of frame- and clip-level evidential opinions to capture holistic uncertainty, we adopt uncertainty-weighted fusion via Dempster’s rule (Eq. (16)). In contrast, UDPF is a hard rule designed to resolve cross-branch conflicts during query categorization, enabling explicit assignments required for subsequent Query Label Calibration with category-specific objectives. We also explore using continuous holistic uncertainty for query categorization. However, the results (dataset: Charades) below show that UDPF performs better.
>
> || R@1|R@5|R@10|R@100|SumR|
> |----------|----|----|----|-----|----|
> |UDPF|2.3|9.5|15.2|53.6|80.6|
> |continuous|2.2|9.1|14.9|53.3|79.5|
>
> >  Need to add a limitations discussion .
>
> **A7**: Thank you for pointing this out. Please refer to our response to `Reviewer S9zY’s W4`.

---

> > ### Author Rebuttal · Reviewer_sJ65 · 2026-04-01
> >
> > I think his response answered my question very well. And I will change the rating to 4.

---

> > > ### Author Response · Authors · 2026-04-01
> > >
> > > We sincerely appreciate your recognition of our paper. We are truly encouraged to know that your concerns have been fully addressed and that you have raised your score to 4. We are deeply grateful for your time, thoughtful feedback, and continued support.

---

### Official Review · Reviewer_M8Q3 · 2026-03-12

**Soundness:** 4
**Presentation:** 3
**Significance:** 3
**Originality:** 3
**Overall Recommendation:** 5
**Confidence:** 3

**Summary:**

This paper proposes a hierarchical evidence learning framework named Holmes, specifically designed to address partially relevant video retrieval. Concretely, the authors transform similarity scores computed by the model into “evidence” and model them mathematically using a Dirichlet distribution. To address the issue of sparse supervision in traditional multiple instance learning, where only the single most relevant segment is emphasized, the authors further introduces a Flexible Optimal Transport (FOT) mechanism. Overall, I believe the paper may offer some novelty, but I would like to further discuss with the authors before determining my final score.

**Compliance With Llm Reviewing Policy:**

Affirmed.

**Final Justification:**

I appreciate that the authors for solving my concerns. I believe this paper offers enough insights to get a positive score.

**Key Questions For Authors:**

1. By introducing the Digamma function computation in the Dirichlet distribution and the optimal transport procedur, how much additional computational time and memory overhead does this method incur during both training and inference, compared to traditional approaches based on the InfoNCE loss?
2. If a query is provided that does not exist anywhere in the entire video corpus, for example, containing completely out-of-domain vocabulary, can Holmes’ epistemic uncertainty mechanism effectively reject retrieval, or will it still produce a confident but incorrect match?

**Limitations:**

Please see above.

**Strengths And Weaknesses:**

**Strengths**:
1. The overall design is interesting. This paper is the first to systematically introduce Dempster–Shafer evidence theory and subjective logic into the domain of untrimmed video retrieval, transforming black-box similarity scores into interpretable probabilities of probabilities distributions.
2. The proposed three principles are intuitive. Through a well-formulated mathematical framework, the method effectively distinguishes between epistemic uncertainty arising from model insufficiency and aleatoric uncertainty caused by inherent data ambiguity.
3. When addressing intra-video segment alignment, the method it introduces the dustbin concept from optimal transport to absorb redundant frames, which is a reasonable design.

**Weaknesses**:
1. The main concern lies in the engineering complexity. The framework integrates multiple highly complex components, including dual-scale feature extraction, evidence computation, triple-threshold decision rules, conflict fusion strategies, and optimal transport. This results in substantial implementation difficulty and increases the risk that modifying one component will affect the entire system. Moreover, such complexity is not conducive to scaling up. In my view, the overall design lacks elegance.
2. Following the above concern, although the authors provide robustness analysis in the appendix, the system introduces a large number of hyperparameters. When applied to a dataset with significantly different distribution characteristics, the cost of tuning these parameters could be substantial. Moreover, when resolving conflicts between the frame-level and segment-level branches, the authors manually define an uncertainty-dominance order. Such a hard-coded rule based on human intuition may not hold in certain edge cases, potentially limiting the method’s generalization.

---

> ### Author Rebuttal · Authors · 2026-03-28
>
> Dear Reviewer `M8Q3`,
>
> We sincerely thank you for your constructive feedback. We are encouraged that you find Holmes **interesting**, with a **reasonable design**, **intuitive three-fold principles**, and a **well-formulated mathematical framework**. Below, we carefully address your concerns. *If you find them satisfactory, we would appreciate your consideration in updating your scores*.
>
> > **W1**: The overall design lacks elegance.
>
> **A1**:
> - We would like to clarify that the complexity largely arises from the widely adopted dual-scale (frame- and clip-scale) retrieval paradigm in PRVR. Following prior work, we employ dual-scale feature extraction, where evidence is computed at each scale and query types are determined via triple-threshold decision rules. To resolve two-scale inconsistencies, the conflict fusion strategy is introduced. In addition, the MIL paradigm in PRVR often results in sparse supervision. We use optimal transport to provide denser supervisory signals. Each component is well-motivated and necessary. Extensive experiments also validate their effectiveness.
> -  Across Tables 1 and 3, Holmes remains competitive with or even surpasses HLFormer despite removing several components, showing the stability of the entire system.
> -  During inference, Holmes employs standard feature extraction and simple similarity computation, without requiring triple-threshold decision rules or optimal transport, ensuring elegance. Furthermore, the complete code will be released to enable easy implementation.
> - Notably, most retrieval tasks adopt a single-scale paradigm, which is simpler and does not require conflict fusion. We show Holmes’s strong performance  under a single-scale setting (e.g. clip-scale only) in the table below (dataset: TVR). Users can flexibly select appropriate retrieval paradigms and components, allowing  low-cost scaling up.
>
> ||R@1|R@5|R@10|R@100|SumR|
> |----------|-----|-----|------|-------|------|
> |Holmes|17.1|38.8|50.6|87.6|194.1|
> |HLFormer|11.4|30.5|41.8| 82.4|166.1|
>
> > **W2-1**: The cost of hyperparameter tuning.
>
> **A2**:  As mentioned, we provide robustness analysis on the TVR dataset in Appendix A.2 and further report results below (dataset: Charades. boldface denotes default values). As shown, the parameters exhibit similar robustness to TVR within a reasonable range, regardless of domain-specific variations. This shows that excessive fine-tuning is not required for different datasets. In Holmes, these hyperparameters are fixed across datasets without dataset-specific tuning.
>
> |SumR|78.8|80.0|80.4|80.6|80.9|80.7|
> |--------|------|------|------|------|------|------|
> |$\tau$|0.04|0.06|0.08| **0.1**|0.15|0.2|
>
> |SumR|80.0|80.6|80.8|80.5|79.6|79.0|
> |--------|------|------|------|------|------|------|
> |$\beta$|0.2| **0.3**|0.4|0.5|0.6|0.7|
>
> |SumR|80.9|80.6|80.4|80.0|79.5|79.2|
> |------|------|------|------|------|------|------|
> |z| 0.2| **0.3**|0.4|0.5|0.6|0.7|
>
> > **W2-2**: The effectiveness of the uncertainty-dominance partition fusion (UDPF).
>
> **A3**:  The UDPF is motivated by the varying semantic reliability across query types. Although deterministic, ablations in Table 3 show it outperforms single-branch baselines and an opposing “confidence dominance” strategy. We also try an adaptive routing method that feeds the query embedding and two-scale three-fold principles into an MLP to learn branch weighting; however, results (dataset: Charades) below show modest performance.
>
> ||R@1|R@5|R@10|R@100|SumR|
> |--------|-----|-----|------|-------|------|
> |UDPF|2.3|9.5|15.2|53.6|80.6|
> |Router|2.2|9.0|14.5|53.1|78.8|
>
> > **Q1**:  The additional computational time and memory overhead.
>
> **A4**:  Because the Digamma function and optimal transport are solely employed in constructing the evidential learning objectives ($L_{inter}$ and $L_{intra}$), they introduce zero computational or memory overhead during inference. At inference time, Holmes only additionally computes uncertainty as defined in Eqs. (3–5), which is highly lightweight. The training overhead is also minimal. These are  confirmed by the results (dataset:  Charades) below.
>
> ||Train time/epoch (ms)|Train memory (MB)|Inference time/epoch (ms)| Inference memory (MB)|
> |--------------|------------------------|-------------------|---------------------------|-----------------------|
> |infonce loss only|23224|8501|5265|2672|
> |full loss|23329|8518|5268|2674 |
>
> > **Q2**: How does Holmes handle out-of-domain queries?
>
> **A5**:  Evidential learning and epistemic uncertainty naturally address out-of-domain (OOD) scenarios. During retrieval, OOD queries typically yield uniformly low similarity scores across all candidates, resulting in low evidence. Holmes still produces a retrieval list, but with high epistemic uncertainty, indicating low confidence. A simple thresholding strategy enables OOD rejection: set the threshold at the top 5% epistemic uncertainty on in-domain queries and reject retrieval when it is exceeded at inference.

---

> > ### Author Rebuttal · Reviewer_M8Q3 · 2026-04-01
> >
> > I appreciate that the authors for solving my concerns. I believe this paper offers enough insights to get a positive score.

---

> > > ### Author Response · Authors · 2026-04-01
> > >
> > > We sincerely appreciate your recognition of our paper. We are truly encouraged to know that your concerns have been fully addressed and that you have raised your score to 5. We are deeply grateful for your time, thoughtful feedback, and continued support.

---

### Decision · Program_Chairs · 2026-04-30

**Decision:**

Accept (regular)

**Comment:**

This paper proposes Holmes, a hierarchical evidential learning framework for partially relevant video retrieval, modeling uncertainty at both inter- and intra-video levels. Reviewers found the problem well-motivated, with strong empirical results and comprehensive ablations. Most concerns regarding efficiency, hyperparameters, and design choices were addressed during rebuttal, leading to improved ratings.

One reviewer raised concerns about novelty and the reliability of the learned evidence, noting that parts of the method build on existing evidential learning techniques and involve heuristic design choices. While these concerns are valid, they are partially mitigated by the coherent integration of components and consistent performance gains across benchmarks.

Overall, the work is technically solid and provides a meaningful contribution to uncertainty-aware PRVR. I recommend accept. The authors are encouraged to further clarify novelty and strengthen the discussion on uncertainty reliability in the final version.